# WHERE DO LARGE VISION-LANGUAGE MODELS LOOK AT WHEN ANSWERING QUESTIONS?

## ABSTRACT

Large Vision-Language Models (LVLMs) have shown promising performance in vision-language understanding and reasoning tasks. However, whether they truly understand the input image remain underexplored. A fundamental question arises: to what extent do LVLMs rely on visual input, and which image regions contribute to their responses? It is non-trivial to interpret the free-form generation of LVLMs due to their complicated visual architecture (*e.g.,* multiple encoders and multi-resolution) and variable-length outputs. In this paper, we extend existing heatmap visualization methods for classification tasks to support LVLMs for open-ended visual question answering. We propose a method to select visually relevant tokens that reflect the relevance between generated responses and the input image. Furthermore, we conduct a comprehensive analysis of state-of-the-art LVLMs on benchmarks designed to require visual information to answer. Our findings offer several insights into LVLM behavior, including the relationship between focus region and answer correctness, differences in visual attention across architectures, and the impact of LLM scale on visual understanding. The code and data will be released.

## 1 INTRODUCTION

The emerging Large Vision-Language Models (LVLM) (Li et al., 2024a; Bai et al., 2023; Team et al., 2023; Dai et al., 2023) have exhibited strong visual instruction following abilities and achieved remarkable performance on multimodal tasks, such as Visual Question Answering (VQA) (Antol et al., 2015). Despite different design and implementation details, most LVLMs follow the representative visual instruction tuning (Liu et al., 2024b) paradigm to align the visual features from pre-trained vison encoders (Radford et al., 2021; Oquab et al., 2024) to a pre-trained LLM (Touvron et al., 2023; Zheng et al., 2023). This enables LVLMs to incorporate visual understanding while retaining the rich knowledge and reasoning abilities of LLMs.

However, the underlying mechanisms behind the visual understanding capabilities of LVLMs remain unclear. Beyond evaluating model performance on various benchmarks, it is crucial to interpret where the LVLM focuses on when generating responses, as it can provide insights into why an answer is correct or incorrect and facilitate targeted improvements for multimodal tasks. For instance, as shown in Figure 1, an LVLM may attend to the correct region but still misinterpret the content (*e.g.,* the top-right example), fail to locate the relevant region entirely (*e.g.,* bottom-right), or even produce correct answers based on irrelevant regions (*e.g.,* bottom-left), which can lead to poor generalization.

Before the era of LVLMs, a popular way to interpret visual models is to derive a saliency heatmap of the input image, representing the relevance of the image regions to the output (Selvaraju et al., 2017; Chefer et al., 2021b; Barkan et al., 2023). Despite the rapid growth of LVLM research, little effort has been made to interpret LVLMs. Existing works (Ben Melech Stan et al., 2024; Giulivi & Boracchi, 2024) often explain single-label outputs or individual tokens within a sentence. However, LVLMs generate open-ended responses consisting of multiple tokens with variable lengths, requiring a holistic interpretation of the entire output rather than isolated components. Interpreting open-ended responses of LVLMs has several challenges, (1) Vision-Language Interaction: LVLMs involve intricate interactions between vision and language modalities, and often exhibit strong bias towards language priors. Hence it is hard to determine the contribution of each modality to the response. (2) Autoregressive Generation: Unlike classification models, LVLMs autoregressively generate free-form text, making it difficult to interpret the model behavior considering the entire output. (3) Model

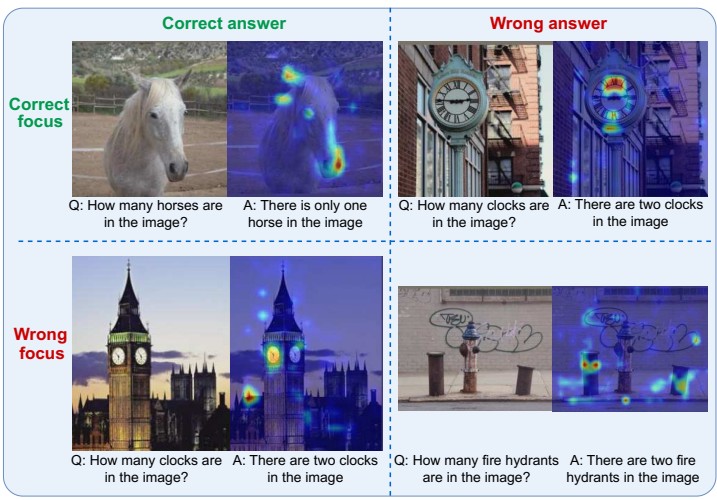

Figure 1: Focus regions of LLaVA-1.5 when answering counting questions. The model may correctly focus on the relevant region and produce the correct answer (top left), or it may fail despite attending to the right region due to misinterpretation (top right). In some cases, incorrect focus leads to wrong answers (bottom right), while occasionally, the model answers correctly despite attending to irrelevant regions (bottom left), highlighting challenges in visual grounding and generalization.

Architecture Complexity: Current LVLMs often use multi-resolution or multi-encoder architectures, making it unreliable to align the features across layers to specific spatial regions in the image.

To address these issues, we propose a method that enables model interpretation for LVLMs and open-ended responses. It reveals the significant image regions that lead to the generated response, providing insights into the reasoning process. To the best of our knowledge, this is the first heatmap visualization method that applies to any LVLM structure and generates a global interpretation for open-ended responses. Using this method, we conduct in-depth studies of state-of-the-art LVLMs on benchmarks designed to evaluate the visual understanding, and obtain several insights related to the model behaviors. (1) LVLMs **do** look at images more often than not on vision-centric datasets: our experiment shows that in more than $75\%$ of the cases removing visual information significantly reduces the answer probability; (2) Impact of Vision Architecture: Different vision architectures could lead to different attention patterns and mechanisms of visual decision-making. But in most cases, the focus regions are reflected in the answers, even when the answer is wrong; (3) Different LLMs or simply increasing the LLM size do not significantly alter the visual attention behavior of the model. Our main contributions are:

- We extended existing visual interpretation methods designed for image classification to interpret LVLMs with free-form text output.

- We proposed to extract visually relevant tokens from the open-ended responses, which are representative of the vision-related parts of the variable-length responses.

- We conducted in-depth studies of LVLM visual behavior, analyzing how different models attend to image regions when answering visual-related questions.

## 2 RELATED WORK

**Visual Instruction Tuning.** Benefiting from the advancement of LLM, current LVLMs often seek to equip LLMs with visual understanding capabilities, which is achieved by visual instruction tuning (Liu et al., 2024b). They utilize a modality connector to align the visual embeddings into prompts that the language models can comprehend (Li et al., 2023a;b). Despite the great progress, existing LVLMs still suffer from hallucination (Jiang et al., 2024a; Liu et al., 2023; Zhang et al., 2023; Zhou et al., 2023) due to insufficient visual understanding capabilities. Improving the visual capabilities of LVLMs is crucial to their further improvement and application, and several benchmarks (Tong et al., 2024a;b; Chen et al., 2024) have been proposed to highlight the growing attention to this issue.

**Visual Attention Visualization.** We mainly focus on a popular family of interpretation methods, which aims to highlight the image regions most relevant to the model's output with a heatmap. Heatmap visualization methods include gradient-based approaches (e.g. (Selvaraju et al., 2017; Zhang et al., 2018)) and perturbation-based approaches (Dabkowski & Gal, 2017; Khorram et al., 2021; Qi et al., 2019; Fong & Vedaldi, 2017). Gradient-based methods often backpropagate the output of the model and generate the heatmap using variants of the gradients (Barkan et al., 2023; Chefer et al., 2021a) and they succeeded in some earlier metrics such as the pointing game (Zhang et al., 2018). However, localization metrics such as the pointing game can be gamed by outputting significant boundaries (Samek et al., 2019; Fuxin et al., 2021), and most gradient-based methods will output the same heatmap even with randomized networks, hence cannot pass the sanity check (Adebayo et al., 2018). Perturbation-based methods deduce region importance by introducing perturbations and analyzing their impact on model output. They often optimize for a mask to the input image to identify the most influential regions. Importantly, perturbation-based methods, such as iGOS++ satisfy the sanity check (Khorram et al., 2021) and do not suffer from the impossibility theorems that would apply to gradient-based algorithms (Bilodeau et al., 2024). Another line of work aligns the model attention with human attention (Chen et al., 2021; Das et al., 2016; Sood et al., 2023) for better grounding. However, deep networks often make decisions with mechanisms different from humans (and even different among different networks (Jiang et al., 2024b) hence cannot be all consistent with humans). Unlike them, we analyze the intrinsic behavior of LVLMs. Traditional model interpretation methods are still limited to interpreting a single output, and can not directly apply to free-form generative models. Besides, current LVLMs often involve complex multi-modal structures with multi-resolution and multiple vision encoders, which makes the gradient-based and transformer interpretation methods (Chefer et al., 2021b; Hao et al., 2021) hard to apply.

**LVLM Interpretation.** Recently the interpretation of LVLMs has drawn increasing interest, as it provides valuable guidance for model development. A pipeline (Ben Melech Stan et al., 2024) has been proposed to visualize the attention of LLaVA (Liu et al., 2024b). Besides, some works visualize the model attention across different layers and either propose to improve the efficiency of LVLMs by pruning the redundant image tokens (Chen et al., 2025; Zhang et al., 2024), or alleviate the hallucination problem through modifying the decoding process (Huang et al., 2024). However, these methods mainly interpret the LVLMs from internal attention. The relevance of different regions on the input image to the output remains under-explored. A recent work (Giulivi & Boracchi, 2024) combines an open-world localization model with the LVLM to generate object localization for the output tokens using the vision embedding, while still limited to object-centric interpretation.

## 3 METHOD

In this section, we first formulate the task and introduce background knowledge on heatmap visualization methods. Next, we propose a visually relevant token selection strategy and generalize the visualization methods to open-ended responses of LVLMs.

### 3.1 PRELIMINARIES

**Task Formulation.** Given an input image $I$ and question $Q$ about the image, an LVLM generates an open-ended answer $a$ in natural language. The answers are generated autoregressively and may vary in length. In autoregressive text generation, words are tokenized and sequentially predicted conditioned on previous tokens. Suppose the answer $a$ consists of $l$ tokens, represented as a sequence $a = [a_1, a_2 \cdots a_l]$. At each step $t$, the model samples the next token $a_t$ according to:

$$a_t \sim P(a_t | a_1, a_2 \cdots a_{t-1}; I, Q) \tag{1}$$

To investigate where the model focuses while generating the answer, we aim to obtain a heatmap $M$ that highlights the importance of each image pixel to the model output. In the next section we introduce the common attention-based and optimization-based heatmap visualization methods.

**Attention-based Heatmap Visualization.** GradCAM (Selvaraju et al., 2017) is a simple and famous approach that obtains gradients at the last layer of convolution. It first computes channel importance by global average pooling the gradient of image classification w.r.t. each output channel (across the image). The importance of the image region is then computed as a weighted average of the activation at an image region across different channels using the channel importance weights.

T-MM Chefer et al. (2021a) aggregates attention from multiple attention layers. For each layer, the attention map is defined as the gradient from the desired category $y_t$ multiplied with the attention

$$\bar{A} = \mathbb{E}_h \left[ ReLU(\frac{\partial y_t}{\partial A} \odot A) \right] \tag{2}$$

where the expectation is taken on the multiple heads of the transformer, and $\odot$ represents element-wise product. Finally, multi-layer transformer attention is computed as $(\mathbf{I} + \bar{A}_L) \dots (\mathbf{I} + \bar{A}_2)\bar{A}_1$ where $\mathbf{I}$ is the identity matrix and $\bar{A}_l$ is the attention map at layer $l$.

IIA Barkan et al. (2023) is an extension of T-MM by incorporating the idea of integrated gradients. With integrated gradients, it computes multiple interpolations between the input image $I$ and a baseline image $B$ as: $rI + (1 - r)B$ with $r = [0, 0.1, \dots, 1]$ and average the gradients of all these images to prevent the gradient of the input image being inaccurate.

**Optimization-based Heatmap Visualization.** iGOS++ Khorram et al. (2021) derive the heatmap $M$ by solving an optimization problem. Suppose the model predicts a score $f$ for the output given $I, Q$, the optimization has two main objectives: *deletion* and *insertion*. *Deletion* progressively removes pixels from $I$ in the order of their heatmap values, aiming to minimize the model's prediction score $f$. *Insertion* starts with a baseline image $\tilde{I}$ without visual information (*e.g.,* fully blurred image) and gradually restores pixels according to their heatmap values, optimizing the heatmap to maximize $f$. Hence the resulting heatmap highlights the most influential regions that contribute to the model's final prediction. The insertion and deletion operation can be denoted as:

$$\Phi(I, \tilde{I}, M) = I \odot M + \tilde{I} \odot (1 - M) \tag{3}$$

where $\odot$ denotes the Hadamard product. Defining deletion and insertion masks $M_x$ and $M_y$ for each objective, the final heatmap $M$ is obtained as their combination: $M = M_x \odot M_y$. The whole objective function is as follows:

$$\min_{M=(M_x, M_y)} f(\Phi(I, \tilde{I}, M_x)) - f(\Phi(I, \tilde{I}, 1 - M_y)) + f(\Phi(I, \tilde{I}, M)) - f(\Phi(I, \tilde{I}, 1 - M)) + g(M)$$

$$\text{where } g(M) = \lambda_1 \|1 - M\|_1 + \lambda_2 BTV(M) \tag{4}$$

$g(M)$ is a regularization term consisting of an $L1$ norm to promote sparsity and a Bilateral Total Variation (BTV) norm (Khorram et al., 2021) to enforce smoothness. The objective minimizes the deletion scores when applying $M_x$ and $M$, and maximizes the insertion scores with $M_y$ and $M$.

However, existing heatmap visualization methods cannot directly apply to the open-ended responses of LVLMs, since these models do not inherently produce a single prediction score. We will discuss how to generalize this approach to open-ended responses in Section 3.2 and our changes to the optimization method in Section 3.3.

## 3.2 VISUALLY RELEVANT TOKEN SELECTION

To extend heatmap visualization methods to free-form text outputs, a representative prediction score $f$ is required for optimization. A simple way is to average all token probabilities. However, empirically with this approach we have observed less consistent explanations and heatmaps that are harder to interpret. We believe this is because autoregressive text generation produces responses of variable length, where token-image correlations may vary significantly. Specifically, token probabilities are influenced by both sentence structure and subword composition. As illustrated in Figure 2, conditional probabilities of most tokens remain largely unchanged under input blur, particularly those that can be inferred from syntax or context (*e.g.*, punctuation, painting, was, by) and the subsequent tokens within a word (*e.g.* ardo, da, V, inci). In contrast, the first token in the sentence and visually relevant tokens often exhibit a notable probability drop, as the first token in the response often decide the subsequent sentence structure, and the visually relevant tokens highly depend on the specific visual input. To this end, we propose to extract the most visually relevant tokens and derive the prediction score from them to achieve a representative interpretation of the whole output.

Due to the autoregressive nature of LVLMs, the probability of generating answer $a$ can be decomposed as the joint probability of its tokens. For simplicity, we omit $I, Q$ and denote the conditional probability of the next token as $P(a_t | a_1 \cdots a_{t-1})$. The probability of the whole sentence is then:

$$P(a|I, Q) = P(a_1)P(a_2|a_1) \cdots P(a_l|a_1 \cdots a_{l-1}) \tag{5}$$

To measure the influence of visual information, we introduce a baseline image $\tilde{I}$ that does not provide any visual information to answer the question. The probability of generating the original answer $a$ given $\tilde{I}$ is denoted as: $P(a|\tilde{I}, Q) = \tilde{P}(a_1) \cdots \tilde{P}(a_l|a_1 \cdots a_{l-1})$, where $\tilde{P}(a_t|a_1 \cdots a_{t-1})$ represents $P(a_t|a_1 \cdots a_{t-1}; \tilde{I}, Q)$. This term can be efficiently computed in a single forward pass by concatenating $Q$ and $a$ as the textual prompt to the model. The difference in confidence is quantified by the log-likelihood ratio Barnett & Bossomaier (2012); Woolf (1957) between the prediction with and without visual information:

$$LLR = \log P(a|I, Q) - \log P(a|\tilde{I}, Q) = \log \prod_t P(a_t|a_1 \cdots a_{t-1}) - \log \prod_t \tilde{P}(a_t|a_1 \cdots a_{t-1})$$

$$= \sum_t \log P(a_t|a_1 \cdots a_{t-1}) - \sum_t \log \tilde{P}(a_t|a_1 \cdots a_{t-1}) := \sum_t LLR_t(a_t) \quad (6)$$

where $LLR_t(a_t) = \log P(a_t|a_1 \cdots a_{t-1}) - \sum_t \log \tilde{P}(a_t|a_1 \cdots a_{t-1})$. Therefore, to identify tokens most influenced by visual information, we apply a threshold to filter those with the highest log-likelihood ratio. The set of crucial tokens $\mathcal{K}$ is selected as $\mathcal{K} = \{a_k| LLR_k > \alpha\}$.

Finally, we define the prediction score $f$ as the cumulative log-likelihood of the crucial tokens. It ensures that only visually relevant parts of the response contribute to the interpretation, filtering out the influence of linguistic structures and leading to a more faithful interpretation of the model's reliance on visual information:

$$f = \sum_{a_k \in \mathcal{K}} \log P(a_k|a_1 \ldots a_{k-1}). \quad (7)$$

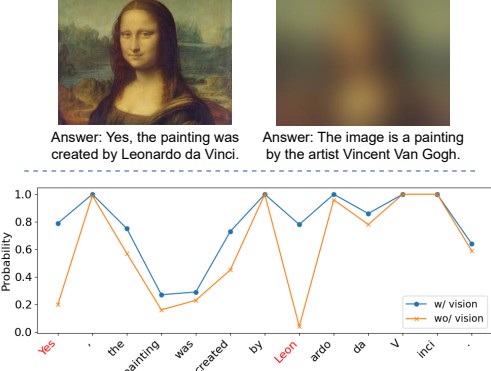

Figure 2: Top: answers generated by LLaVA-1.5 given the original image and fully blurred baseline image. Bottom: conditional probability of the original answer given input image and baseline image. Most tokens in the response are not very dependent on the visual information.

### 3.3 ADAPTATION TO LVLMs

We introduce some additional measures to adapt heatmap visualization approaches to LVLMs with different model architectures.

**Multi-encoder and Multi-resolution.** Current LVLMs often leverage multiple vision encoders (Tong et al., 2024a) or multi-resolution (Li et al., 2024a;b) to enhance visual understanding, which introduces challenges when applying the mask in optimization-based method. In the cases where the input image is processed by multiple vision encoders before integrating their features, we apply a single unified mask to the input image before passing it through all encoders, ensuring consistency across different feature extractors. Multi-resolution methods often crop the input image into variable-sized patches according to the original resolution, where the operation is nondifferentiable and obstructs the optimization. We implement an equivalent differentiable cropping operation (replace pillow and numpy operations with tensor operations), ensuring that the mask undergoes the same transformation as the image patches. This allows us to apply the multi-resolution mask to the corresponding image patches while maintaining differentiability. Besides, for attention-based methods, we only calculate the gradients to the image, eliminating the influence of the textual input. We also use gradient checkpointing to reduce the memory requirement when interpreting large-scale models.

**Improvement on iGOS++.** In practice, the heatmap optimization is non-convex and difficult to converge. Besides, the reasoning process of LVLMs is complicated, with responses often related to multiple image regions, leading to scattered attention maps that are hard to optimize. For simplicity, we directly optimize a single mask $M$ for both deletion and insertion objectives. To further improve stability, we incorporate graduated non-convexity (Hazan et al., 2016) (GNC) to reduce the risk of getting into local optima and the oscillations during optimization. Instead of directly solving a non-convex problem, GNC begins with a convex approximation as a more stable starting point, and gradually introduces non-convexity into the optimization process. Specifically, we add an exponentially decayed $L2$ norm to the objective function, yielding the final formulation:

$$\min_M \ f(\Phi(I, \tilde{I}, M)) - f(\Phi(I, \tilde{I}, 1 - M)) + \lambda_1 \|1 - M\|_1 + \lambda_2 e^{-\gamma t} \|1 - M\|_2 + \lambda_3 BTV(M) \quad (8)$$

and we set $\lambda_1 = 1, \lambda_2 = 0.1, \lambda_3 = 10$ as the default value.

## 4 EXPERIMENT

We now experiment with heatmap visualization approaches extended to interpret open-ended output of LVLMs. First, we evaluate different interpretation methods using quantitative metrics. Next, to gain insights into the visual behavior of state-of-the-art open-source LVLMs, we conduct quantitative and qualitative experiments focusing on the following key research questions. **Q1**: Do LVLMs rely on the input image when answering visual questions? **Q2**: Where do different LVLMs attend when generating variable-length responses? **Q3**: What is the relationship between answer correctness and focus region? **Q4**: How do the vision encoder and LLM components influence visual behavior?

### 4.1 EXPERIMENT SETTINGS

We evaluate LVLMs that employ representative strategies to improve visual instruction following capabilities: LLaVA-1.5 (Liu et al., 2024a) leverages a fully connected cross-modal adapter and incorporates academic-related data (Goyal et al., 2017) to enhance visual instruction tuning. LLaVA-OneVision (LLaVA-OV) (Li et al., 2024a) employs multi-resolution input images, hence captures finer image details. Cambrian (Tong et al., 2024a) proposes a Spatial Vision Aggregator to integrate visual features from multiple encoders.

We select recent datasets that target at evaluating the visual instruction following capabilities of LVLMs: MMStar (Chen et al., 2024) contains 1,500 human-reviewed vision-dependent questions that most LVLMs fail to answer correctly without visual input. CV-Bench (Tong et al., 2024a) constructs a "vision-centric" benchmark with 2,638 manually verified image-related questions. MMVP (Tong et al., 2024b) selects a subset of 300 questions with the images that CLIP (Radford et al., 2021) fails to distinguish. In Appendix A.3 we compare the peak GPU memory usage and the average runtime per sample of our method adapted to different heatmap visualization methods.

### 4.2 STATISTICAL ANALYSIS

To investigate **Q1** (whether LVLMs rely on visual input), we first conduct a statistical analysis comparing the models' responses with and without visual information. Using the visual relevance metric in Eq. 6, we compute the probability of an answer given the original image

Table 1: Percentage (%) of samples that the answer probability decreases by less than 30% / 30%-70% without visual information.

|  | CV-Bench | MMStar | MMVP |
|---|---|---|---|
| LLaVA-1.5 | 8.9 / 18.5 | 7.5 / 10.4 | 25.3 / 16.3 |
| LLaVA-OV | 13.6 / 20.2 | 4.7 / 5.7 | 16.3 / 15.0 |
| Cambrian | 1.1 / 2.3 | 4.2 / 5.0 | 3.3 / 7.3 |

versus a fully blurred image. Table 1 reports the percentage of samples where the answer probability decreases by less than 30% or between 30%-70% without visual information. The results indicate that most responses are affected by the image to varying degrees. Notably, LLaVA-1.5 exhibits lower reliance on visual input on MMStar and MMVP; with 25.3% MMVP samples showing a probability drop of less than 30% when the image is blurred. The responses of Cambrian are more influenced by the visual contents; with answer probabilities decreasing by more than 70% for $\sim$ 90% of samples across datasets. Most compared models have lower density of small probability drops on MMStar, suggesting they rely more heavily on image when answering MMStar questions.

### 4.3 COMPARISON OF VISUALIZATION METHODS

Our proposed token selection method can be applied to various interpretation methods and extend them to open-ended responses of LVLMs. However, some interpretation techniques are not well-suited for LVLMs. We applied our token selection method to the gradient-based method Grad-CAM (Selvaraju et al., 2017), transformers interpretation methods T-MM (Chefer et al., 2021a) and IIA (Barkan et al., 2023), and our improved optimization-based method based on IGOS++ (Khorram et al., 2021).

**Evaluation Metric.** We follow the commonly used *deletion* and *insertion* (Petsiuk, 2018) scores to assess the heatmaps. Deletion removes pixels from the original image in descending order of their heatmap values, and calculates the output scores given the intermediate image to derive a deletion curve. The deletion score is the area under the curve (AUC). Similarly, the insertion score measures how quickly the output score increases when adding pixels to a baseline image. Lower deletion score and higher insertion score indicate heatmap that better reflects areas the model attends to. For fair comparison across models with varying prediction score distributions, we normalize the scores according to those of the original image and baseline image, following Jiang et al. (2024b).

**Data Selection.** For subsequent studies, we filter out the samples where models' answers are largely independent of the image, as it is infeasible to study where the model attends if it does not need the

Table 2: Quantitative comparison of interpretation methods in terms of Deletion score (lower is better), Insertion score (higher is better) on the filtered dataset using different LVLMs. The bold numbers denote the best results for each model and dataset.

| LVLM | Method | MMVP Del ↓ | MMVP Ins ↑ | MMStar Del ↓ | MMStar Ins ↑ | CV-Bench Del ↓ | CV-Bench Ins ↑ | LLaVA-Bench Del ↓ | LLaVA-Bench Ins ↑ |
|---|---|---|---|---|---|---|---|---|---|
| LLaVA-1.5-7b | Grad-CAM | 0.679 | 0.441 | 0.689 | 0.372 | 0.651 | 0.587 | 0.685 | 0.379 |
| | T-MM | 0.778 | 0.418 | 0.869 | 0.283 | 0.869 | 0.461 | 0.808 | 0.378 |
| | IIA | 0.401 | 0.805 | 0.333 | 0.870 | 0.457 | 0.884 | 0.371 | 0.824 |
| | iGOS++ | **0.366** | **0.811** | **0.292** | **0.953** | **0.402** | **0.965** | **0.358** | **0.864** |
| LLaVA-OV-7b | Grad-CAM | 0.341 | 0.501 | 0.406 | 0.550 | 0.443 | 0.562 | 0.334 | 0.549 |
| | T-MM | 0.528 | 0.589 | 0.575 | 0.620 | 0.594 | 0.630 | 0.589 | 0.655 |
| | IIA | 0.576 | 0.551 | 0.621 | 0.578 | 0.660 | 0.553 | 0.541 | 0.612 |
| | iGOS++ | **0.305** | **0.778** | **0.317** | **0.870** | **0.295** | **0.924** | **0.301** | **0.803** |
| Cambrian-8b | Grad-CAM | 0.416 | 0.513 | 0.391 | 0.656 | 0.471 | 0.599 | 0.452 | 0.582 |
| | iGOS++ | **0.375** | **0.657** | **0.334** | **0.860** | **0.340** | **0.849** | **0.372** | **0.786** |

image. We keep the samples where all models have clear probability differences with and without visual information, remaining 35% samples from MMVP, 47% of MMStar samples, and 34% of CV-Bench. The original datasets are multiple-choice questions, so we remove the choices and instructions to relax them into open-ended questions. Additionally, we evaluate on LLaVA-Bench (Liu et al., 2024b) designed for open-ended VQA.

**Experiment Results.** With the selected data, we generate heatmaps using different interpretation methods and compare their deletion and insertion scores in Table 2. As Cambrian uses multiple vision encoders including non-transformer-based models, T-MM and IIA are not applicable. iGOS++ consistently outperforms all other methods, with the lowest deletion scores and highest insertion scores across datasets and mod-

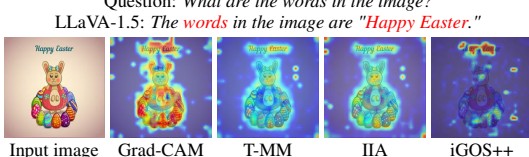

Question: *What are the words in the image?*
LLaVA-1.5: *The words in the image are "Happy Easter."*

Input image    Grad-CAM    T-MM    IIA    iGOS++

Figure 3: Qualitative comparison of different explanation methods. The tokens in red denote the selected crucial tokens.

els. Qualitative heatmap visualizations shown in Figure 3 also show that iGOS++ generates more precise heatmap than attention-based methods. Hence, we choose iGOS++ for subsequent analyses.

## 4.4 ABLATION STUDY

We conduct ablation studies to demonstrate the effectiveness of the proposed token selection strategy, as an important contribution of our method is handling long-sentence responses by identifying visually relevant tokens based on the log-likelihood ratio. We compare with (1) Computing the joint probability of the whole

Table 3: Ablation study of token selection strategy. The bold numbers denote the best and underlined denote second-best results.

| | MMVP Del ↓ | MMVP Ins ↑ | MMStar Del ↓ | MMStar Ins ↑ | CV-Bench Del ↓ | CV-Bench Ins ↑ |
|---|---|---|---|---|---|---|
| Proposed | **0.366** | **0.811** | 0.292 | **0.953** | **0.402** | 0.965 |
| Joint prob. | 0.403 | 0.784 | **0.286** | 0.906 | 0.409 | **0.967** |
| Keywords | 0.367 | 0.809 | 0.302 | 0.931 | **0.402** | 0.899 |

sentence. (2) Detecting keywords using off-the-shelf tagging method (Campos et al., 2018; 2020). As shown in Table 3, our proposed token selection approach leads to the most relevant heatmaps across all datasets, improving over *joint probability* by 2.7% on MMVP and 4.7% on MMStar, and improving over *keywords* by 6.6% on CV-Bench. We also include the ablation studies of the GNC norm in Eq. 8, token selection threshold $\alpha$ and baseline image selection for optimization-based method in Appendix A.1.

## 4.5 FOCUS REGION ANALYSIS

**Heatmap Visualization.** We present qualitative results in Figure 4 to address **Q2** about the focus region of different LVLMs when generating outputs. We highlight the visually relevant tokens in red. The results lead to several observations: (1) Cambrian reveals more compositional (Jiang et al., 2024b) image understanding, which means it tends to jointly consider the entire image and may include more comprehensive information in its responses (*e.g.*, in example (b) Cambrian provides more information about the mountainous landscape). In contrast, LLaVA-OV shows more disjunctive behaviors and often focuses on specific regions. This aligns well with their respective architectures: Cambrian aggregates multiple vision encoders to extract broader visual information, while LLaVA-OV adopts a multi-resolution strategy to extract detailed features. (2) The responses of the models

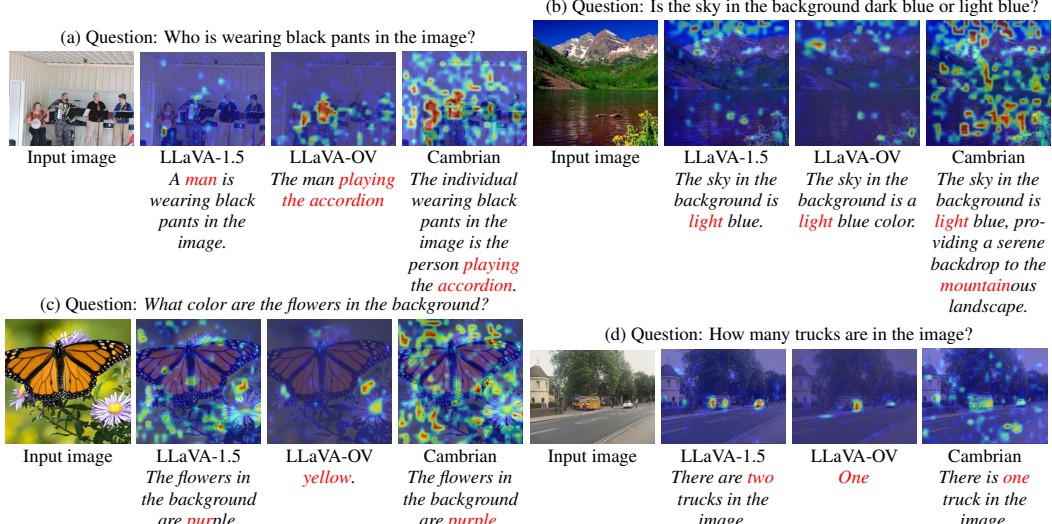

Figure 4: Comparison of the generated response and focus region of different LVLMs. Tokens in red are the selected visual relevant tokens.

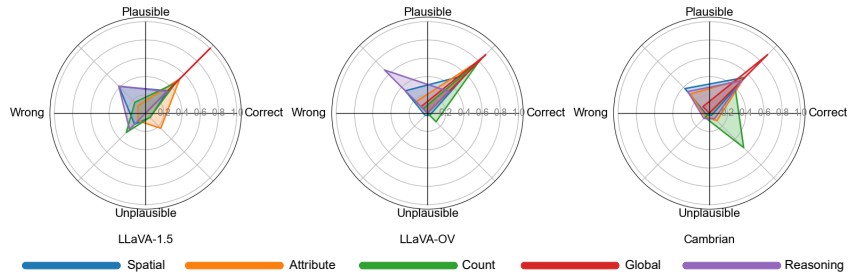

Figure 5: Answer correctness and focus region plausibility across four quadrants. Each color stands for a different question category, including spatial, attribute, counting, global context, and reasoning.

may include different contents since they attend to different image regions. For instance, in example (a) LLaVA-1.5 only attends to the man's face but LLaVA-OV also looks at the accordion he plays, hence has "playing the accordion" in its answer. (3) When LVLMs give incorrect answers, their focus regions could reveal the underlying cause. In example (c), LLaVA-OV locates the pistil instead of the petal hence answers yellow for the color of the flower. In example (d), LLaVA-1.5 mistakenly attends to the sedan and counts it as a truck. More visualization results are included in the supplemental.

**Answer vs. Focus Region.** To address **Q3**, we conduct a subjective analysis on the relationship between answer correctness and focus region plausibility (*i.e.,* whether the focus region aligns with human intuition). We randomly select 100 samples for and categorize them into spatial, attribute, counting, global context, and reasoning questions. The results are summarized in Figure 5, where we classify model behaviors into four quadrants and calculate the percentage of samples in each quadrant. The key observations include: (1) All models tend to provide correct answers with plausible focus regions when answering global context questions. Conversely, models often fail to answer reasoning and spatial questions even when attending to the right regions. (2) LLaVA-1.5 has lower chance to locate the most relevant regions. LLaVA-OV has better focus plausibility on most question types, though it does not necessarily lead to better accuracy. Cambrian performs well on counting questions, yet often attends to regions that do not align with human intuition. These findings suggest that focus region plausibility does not always correlate with answer correctness. In Appendix A.2, we provide additional experiments to compare the model focus and human attention.

**Influence of Vision Architecture and LLM.** Since LVLMs consist of vision encoders and LLM, we investigate their impact on the focus region, respectively. To study the influence of LLM scale, we compare LLaVA-OV 0.5b, 7b, 72b and Cambrian 3b, 8b, 13b. For models with the same LLM but different vision architectures, we further include Mini-Gemini Li et al. (2024b) as they provide

Table 4: Comparison of models with different LLM scales and vision architectures. It can be seen that the scores are consistent within the same model family. For Cambrian models, they use a combination of 4 vision encoders: CLIP ViT-L (Radford et al., 2021), SigLIP ViT (Zhai et al., 2023), OpenCLIP ConvNeXt-XXL (Ilharco et al., 2021) and DINOv2 ViT-L (Oquab et al., 2024). For Mini-Gemini models, *HD* denotes high resolution.

| Model | LLM | Vision Encoder | Del↓ | Ins↑ |
|---|---|---|---|---|
| Comparison of models with different LLMs | | | | |
| LLaVA-OV-0.5b | Qwen2-0.5b | SigLIP | 0.313 | 0.792 |
| LLaVA-OV-7b | Qwen2-7b | SigLIP | 0.305 | 0.778 |
| LLaVA-OV-72b | Qwen2-72b | SigLIP | 0.304 | 0.754 |
| Cambrian-3b | Phi-3-3.8B | Multi-encoder | 0.424 | 0.664 |
| Cambrian-8b | LLaMA3-8B | Multi-encoder | 0.375 | 0.657 |
| Cambrian-13b | Vicuna1.5-13B | Multi-encoder | 0.415 | 0.692 |
| Comparison of models with different vision architectures | | | | |
| LLaVA-1.5-7b | Vicuna1.5-7B | CLIP | 0.366 | 0.811 |
| Mini-Gemini-7b | Vicuna1.5-7B | CLIP-L | 0.474 | 0.669 |
| Mini-Gemini-7b-HD | Vicuna1.5-7B | ConvNext-L | 0.478 | 0.661 |
| Cambrian-13b | Vicuna1.5-13B | Multi-encoder | 0.415 | 0.692 |
| Mini-Gemini-13b | Vicuna1.5-13B | CLIP-L | 0.473 | 0.674 |
| Mini-Gemini-13b-HD | Vicuna1.5-13B | ConvNext-L | 0.471 | 0.671 |

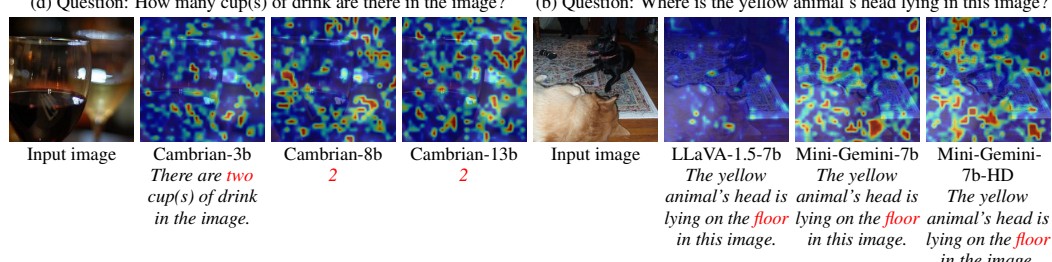

(d) Question: How many cup(s) of drink are there in the image? (b) Question: Where is the yellow animal's head lying in this image?

Input image | Cambrian-3b *There are two cup(s) of drink in the image.* | Cambrian-8b *2* | Cambrian-13b *2* | Input image | LLaVA-1.5-7b *The yellow animal's head is lying on the floor in this image.* | Mini-Gemini-7b *The yellow animal's head is lying on the floor in this image.* | Mini-Gemini-7b-HD *The yellow animal's head is lying on the floor in the image.*

Figure 6: Comparison of models with different LLM scales and vision architectures. The first group compares models with varying LLM sizes while keeping the vision architecture fixed, showing that increasing the LLM scale has minimal impact on visual behavior. The second group compares models with the same LLM but different vision encoders, indicating vision architectures may affect the focus region.

high-resolution models (denoted as HD) with an additional vision encoder. The compared models cover wide range of mainstream LLM structures including Qwen Yang et al. (2024), Phi Abdin et al. (2024), LLaMA Dubey et al. (2024) and Vicuna Zheng et al. (2023). The quantitative results are shown in Table 4.We show qualitative examples in Figure 6. From both quantitative and qualitative results, it can be observed that merely increasing the LLM scale does not essentially change the focus region (p=0.121), despite the differences in response phrasing. In contrast, given the same LLM, varying the vision architecture significantly affects the focus regions (p=0.0008). On the other hand, the LLaVA family shows similar deletion/insertion scores and generally display disjunctive behavior while Cambrian and Mini-Gemini have similar scores and generally display compositional behavior. We show more visualization results in supplementary materials.

## 5 CONCLUSION

In this paper we propose a method to generalize existing visual interpretation methods to support the autoregressive, open-ended responses of LVLMs. We introduce a visually relevant token selection strategy that detects the crucial tokens in variable-length outputs and associates them with specific image regions. With the interpretation method, we conduct a comprehensive analysis of state-of-the-art open-source LVLMs with diverse model structures on visual instruction following benchmarks that require visual information. The experiment results provide several insights into model behaviors, including the relevance of the responses to visual input, relationship between answer correctness and focus region, influence of vision architectures and LLM scales. Despite some limitations discussed in Appendix A.7, these findings emphasize the need for evaluation beyond standard accuracy metrics, offering insights into potential improvements of LVLMs.

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

## A APPENDIX

In Section A.1 we conduct ablation studies on the GNC norm of Eq. 8, token selection threshold $\alpha$ and baseline image selection in the optimization-based heatmap visualization. In Section A.2 we compare the generated model focus region with human attention. In Section A.3 we show the maximum GPU memory usage and average inference time of different heatmap visualization methods extended by our proposed token selection strategy. The experiment results are conducted without LLM inference optimization. We provide additional qualitative examples in Section A.4, A.5, A.6 to better demonstrate the observations in the main paper, comparing different model structures, LLM scales and vision architectures. For each example, we show the generated responses of LVLMs, the detected visually relevant tokens and the corresponding focus regions. We compare the visual behaviors of different model structures and model sizes. In Section A.7 we include a short discussion about the limitations of this work. We also provide the code with guidance to reproduce the experiment results in the supplementary materials.

### A.1 ABLATION STUDIES

In this section, we conduct ablation studies to evaluate the impact of the GNC norm, $\alpha$ for token selection threshold and baseline image selection. We use LLaVA-1.5-7b for the following experiments.

**GNC norm.** To achieve better optimization stability, we introduce graduated non-convexity (GNC) by adding an exponentially decayed L2 norm. Here we conduct parameter studies to evaluate the impact of the weight of the L2 norm. We vary the scale of the L2 norm $\lambda_2$ among [0.0, 0.1, 1.0, 10.0] and compare the deletion and insertion scores on different datasets, as shown in Table 5. It shows that introducing a proper scale of L2 norm can benefit the optimization and obtain better results in terms of the deletion and insertion. Extremely large values of the regularization may disturb the objective function and degrade the performance.

Table 5: Parameter study of the scale of the $L2$ norm in graduated non-convexity. The bolded numbers denote the best results among the compared parameter settings. In the main experiments we select $\lambda_2 = 0.1$.

| $\lambda_2$ | MMVP | | MMStar | | CV-Bench | |
|---|---|---|---|---|---|---|
| | Del↓ | Ins↑ | Del↓ | Ins↑ | Del↓ | Ins↑ |
| 0.0 | 0.381 | 0.794 | 0.295 | **0.953** | **0.398** | **0.965** |
| 0.1 | **0.366** | **0.811** | **0.292** | **0.953** | 0.402 | **0.965** |
| 1.0 | 0.514 | 0.705 | 0.420 | 0.846 | 0.524 | 0.885 |
| 10.0 | 0.513 | 0.735 | 0.412 | 0.858 | 0.520 | 0.894 |

**Token Selection Threshold.** We also compare the interpretation performance given different token selection threshold $\alpha$ and show the results in Table 6. Small threshold $\alpha$ may select too many tokens and introduce noise, while large threshold causes the interpretation to miss relevant content in the output.

Table 6: Ablation study of the token selection threshold $\alpha$ on MMVP dataset. The bolded numbers denote the best results.

| $\alpha$ | Deletion ↓ | Insertion ↑ |
|---|---|---|
| 0.5 | 0.414 | 0.774 |
| **1.0** | **0.366** | **0.811** |
| 2.0 | 0.425 | 0.792 |

**Baseline Image.** In the optimization-based visualization method, the baseline image $\tilde{I}$ should contain minimal visual information (Fong & Vedaldi, 2017), while maintaining a distribution consistent with natural images to avoid introducing adversarial artifacts. We compare different choices of the baseline image in Table 7, including blurred image, all-zero input, and random noise, where we make sure that the average probability of the selected tokens is lower than $5\%$ on the blurred image. Results

indicate that the blurred baseline image achieves the best balance, minimally disturbing the input image distribution.

Table 7: Ablation study of the baseline image selection. The bolded numbers denote the best results.

|  | Del ↓ | Ins ↑ | Del ↓ | Ins ↑ | Del ↓ | Ins ↑ |
|---|---|---|---|---|---|---|
| Blurred | **0.366** | **0.811** | **0.292** | **0.953** | 0.402 | **0.965** |
| Blank | 0.407 | 0.644 | 0.317 | 0.805 | 0.403 | 0.837 |
| Noise | 0.381 | 0.696 | 0.295 | 0.838 | **0.392** | 0.884 |

## A.2 COMPARISON WITH HUMAN ATTENTION

In this paper we mainly investigate the intrinsic behavior of the LVLMs in terms of the focus region when generating the open-ended responses. The model attention is an objective fact which does not need to be aligned with human attention/behavior. However, we conduct experiments to compare the model behavior with human attention to gain deeper insights. We evaluate the LVLMs on VQA-HAT dataset Das et al. (2016), which consists of human visual attention maps over the images in the VQA dataset Antol et al. (2015). In Table 8 we show the soft IOU and rank correlation between the focus region of LLaVA-1.5/LLaVA-OV and human attention labels on VQA-HAT dataset. IOU measures the overlap between the model attention maps and human attention maps, defined as the ratio of their intersection to their union and the range is between 0 and 1. Rank correlation is a statistical measure that quantifies the similarity between the rankings of two variables, commonly used to assess monotonic relationships. The results indicate that the focus regions of LVLMs can significantly differ from human attention, with small IOU and negative rank correlation values.

Table 8: Comparison of model focus region with human attention on VQA dataset, evaluated by the IOU and rank correlation.

|  | IOU | Rank correlation |
|---|---|---|
| LLaVA-1.5 | 0.010 | -0.201±0.003 |
| LLaVA-OV | 0.012 | -0.195±0.004 |

## A.3 COMPUTATION COST

In this section, we conduct experiments to compare the maximum GPU memory usage and average inference time of different heatmap visualization methods extended by our proposed token selection strategy. The results are shown in Table 9. Although the optimization-based method iGOS++ achieves better performance in terms of deletion and insertion scores, it requires the longest inference time, while still in an acceptable range. Using inference optimization methods like vllm Kwon et al. (2023), SGLang Zheng et al. (2024) may further reduce the memory requirement and computation time.

Table 9: Comparison of different visualization methods by maximum GPU memory usage and inference time per sample.

| Method | Max GPU Memory | Time per sample |
|---|---|---|
| iGOS++ | 31 GB | ∼7.0 s |
| Grad-CAM | 20 GB | ∼1.0 s |
| T-MM | 36 GB | ∼2.7 s |
| IIA | 22 GB | ∼4.7 s |

## A.4 COMPARISON OF DIFFERENT MODEL STRUCTURES

In Figure 7, 8, 9 we show more qualitative results, comparing the responses and focus regions of LLaVA-1.5-7b, LLaVA-OV-7b, Cambrian-8b and Mini-Gemini-7b. We mainly categorize the questions into spatial, attribute, counting, global and reasoning questions according to their knowledge types. Figure 7 shows attribute questions that ask about specific information about the objects or

elements in the images. Figure 8 (a) and (b) show counting questions, (c) and (d) spatial relationship questions. Figure 9 (a) and (b) show global questions that require understanding of the whole scene, (c) and (d) show reasoning questions that involve external knowledge and reasoning beyond the visual information to answer. In general, it can be observed that LLaVA-1.5 and LLaVA-OV are more disjunctive, focusing on most relevant regions in the image. In contrast, Mini-Gemini and Cambrian show more compositional visual attention, which means they tend to look at the entire image when answering the questions.

### A.5 COMPARISON OF MODELS WITH DIFFERENT LLM SCALES

With the increasing scale of vision and language models, many recent approaches aim to improve performance by enlarging the model's parameter size. Motivated by this trend, we investigate how model scale—specifically, the scale of LLMs—affects visual focus regions and multimodal behavior. In Figure 10, we compare the responses and corresponding focus region of LLaVA-OV 0.5b, 7b, 72b. In Figure 11, we compare the responses and corresponding focus region of Cambrian 3b, 8b, 13b. In each group of compared models, they have the same vision architecture and multimodal interaction, but different LLM scales. It can be observed that the responses of the models with different LLM scales may have different expressions, but the corresponding focus regions often have similar structures. It indicates that the focus region is not significantly affected by the scale of the LLM. Instead, the LLM scale primarily influences the linguistic expression of the responses.

### A.6 COMPARISON OF MODELS USING THE SAME LLM

In this section we provide more examples to compare the LVLMs with different vision architectures but using the same LLM. In Figure 12 we compare LLaVA-1.5-7b, Mini-Gemini-7b, and Mini-Gemini-7b-HD since they use the same LLM (*i.e.,* Vicuna-1.5-7b). Mini-Gemini-7b-HD provides a high-resolution version that leverages additional vision encoder. Compared with the influence of LLM scale discussed in Section A.5, vision architecture may have more significant influence on the focus region related to the model outputs. It indicates that when aiming to control the visual attention of the LVLMs, we may need to develop targeted design for the vision architecture.

### A.7 LIMITATIONS

In this paper we propose a visually relevant token selection method and extend existing interpretation methods to support open-ended responses of LVLMs with several technical improvements. It illustrates the model generation by deriving a heatmap of the focus region on the image, and can be applied to various model structures with multi-encoder and multi-resolution. In this section, we discuss about the potential limitations of our proposed method. Since the objective function aims to minimize the deletion score (*i.e.*, replacing pixels in the original image with those in the baseline image) and maximize the insertion score (*i.e.*, replacing pixels in the baseline image with those in the original image), the optimization may not be effective when the output scores given the original image and baseline image do not have significant difference. This is also natural that we can not get the focus region if the responses of the model are not highly related to the input image. In such cases, other interpretation methods can be used as complementary.

### A.8 CODE AND REPRODUCTION

We also provide the full source code in the supplementary material. To reproduce all the experiments in this paper, please follow the instructions in *README*.

### A.9 LLM USAGE

We leveraged LLMs to polish the writing of this paper. In particular, LLMs were used to:

1. Check grammar and improve readability.
2. Fix typos.

No LLMs were used for generating ideas, designing methods, or conducting experiments.

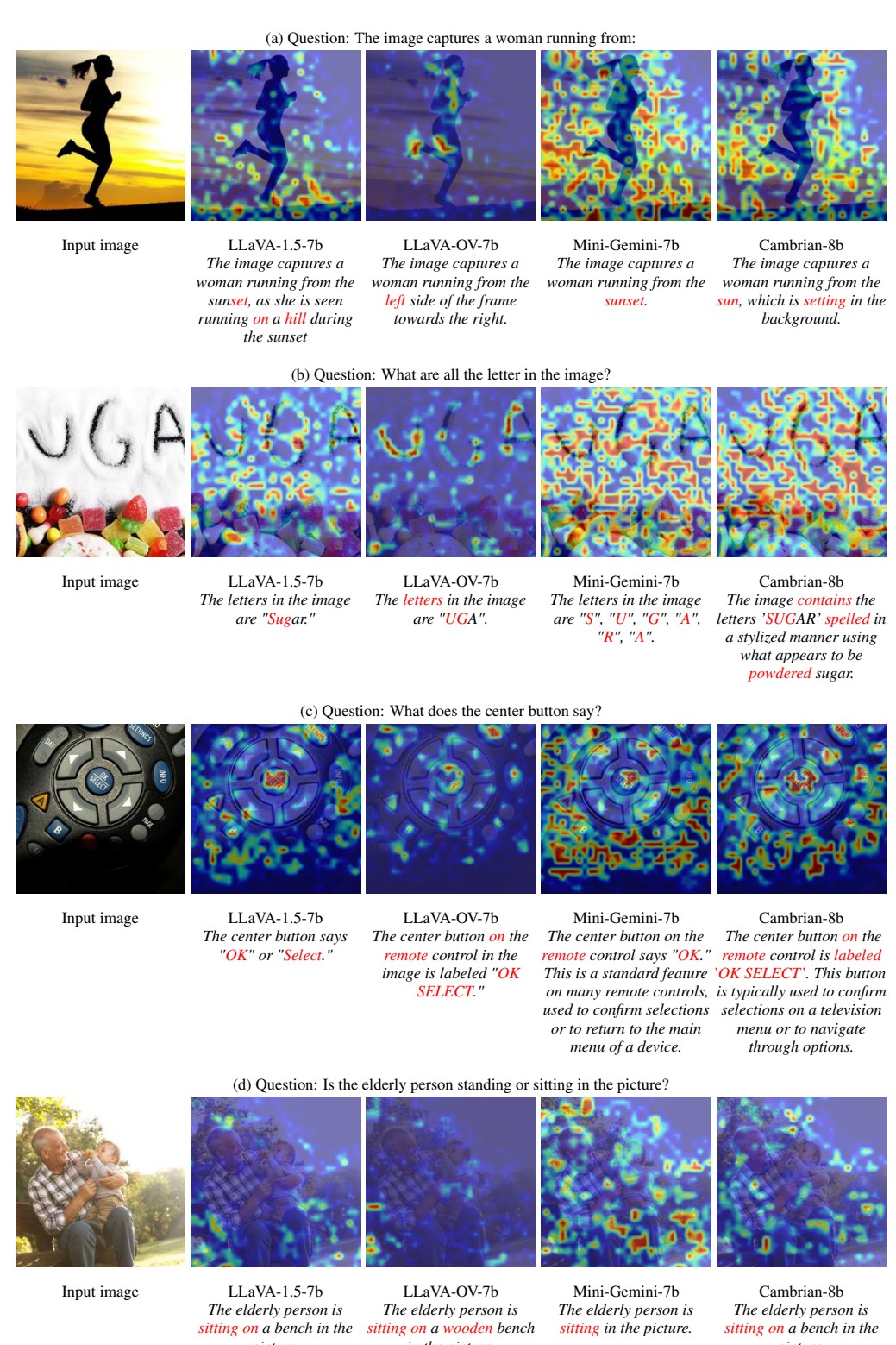

Figure 7: Comparison of the focus region of LVLMs with different model architectures, including LLaVA-1.5-7b, LLaVA-OV-7b, Mini-Gemini-7b and Cambrian-8b. The tokens in red denote the selected visually relevant tokens. The questions ask about specific attributes of the elements in the image.

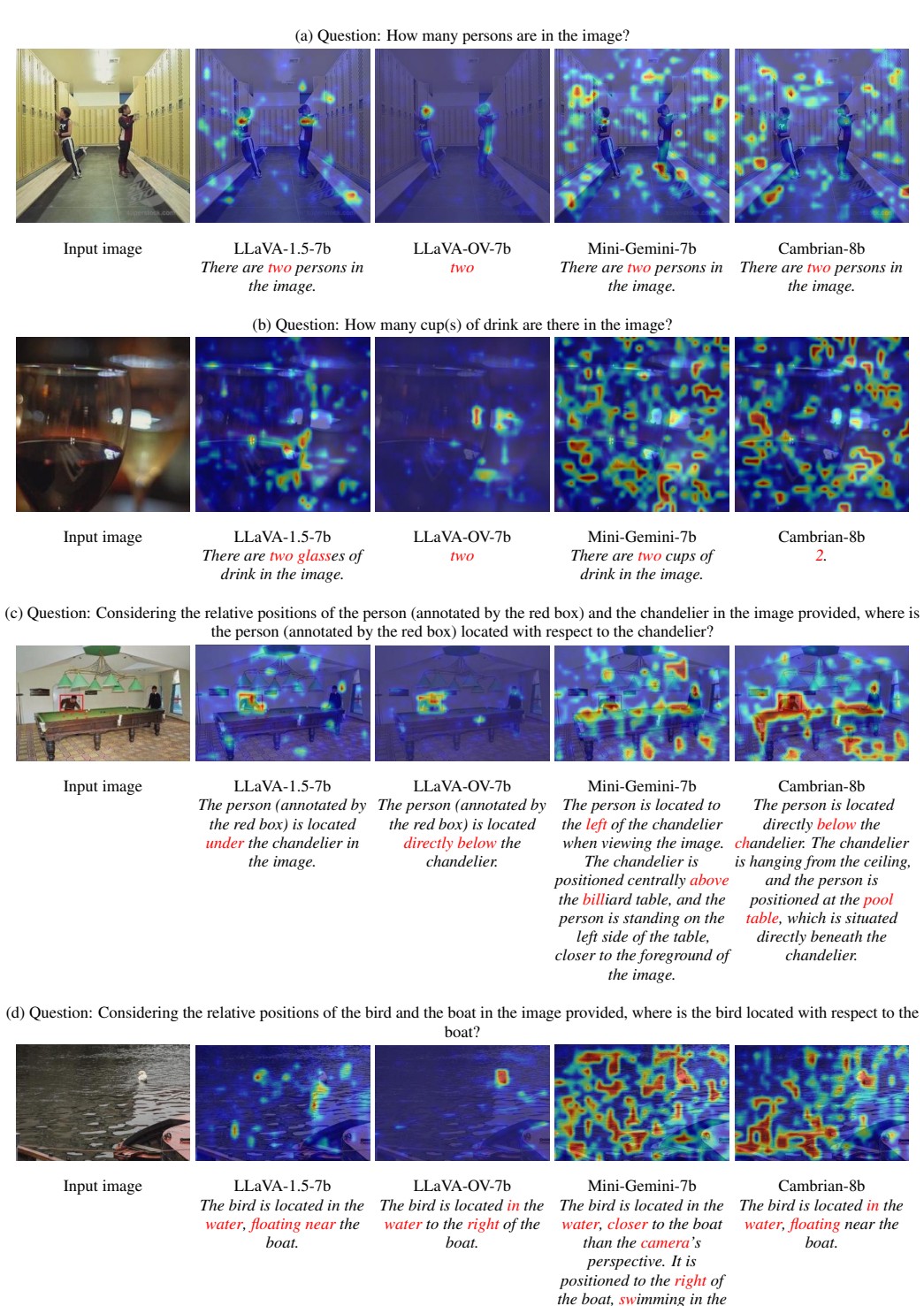

Figure 8: Comparison of the focus region of LVLMs with different model architectures, including LLaVA-1.5-7b, LLaVA-OV-7b, Mini-Gemini-7b and Cambrian-8b. The tokens in red denote the selected visually relevant tokens. The questions in (a) and (b) are counting questions, (c) and (d) are spatial relationship questions.

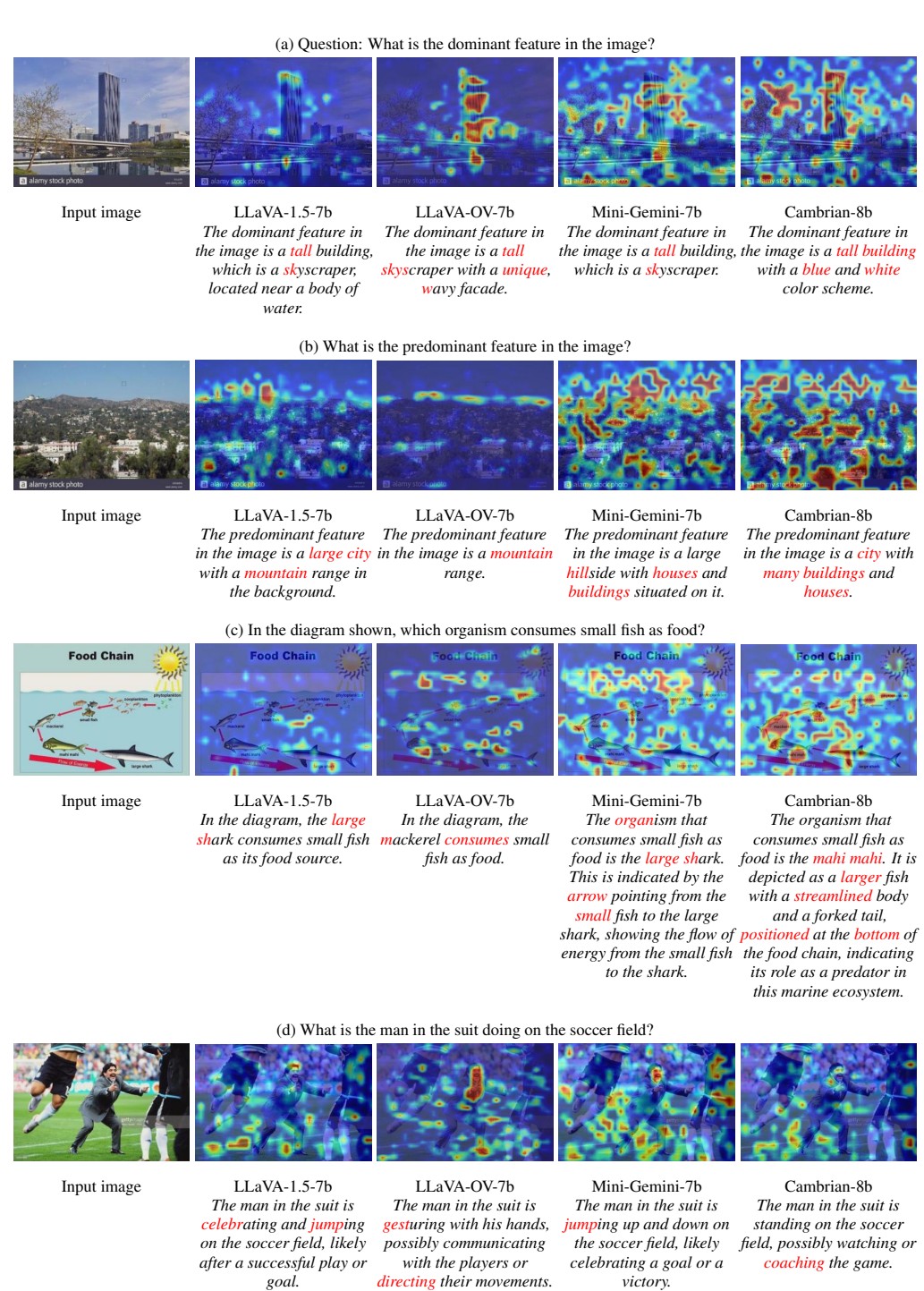

Figure 9: Comparison of the focus region of LVLMs with different model architectures, including LLaVA-1.5-7b, LLaVA-OV-7b, Mini-Gemini-7b and Cambrian-8b. The tokens in red denote the selected visually relevant tokens. The questions in (a) and (b) are global understanding questions, (c) and (d) are reasoning questions.

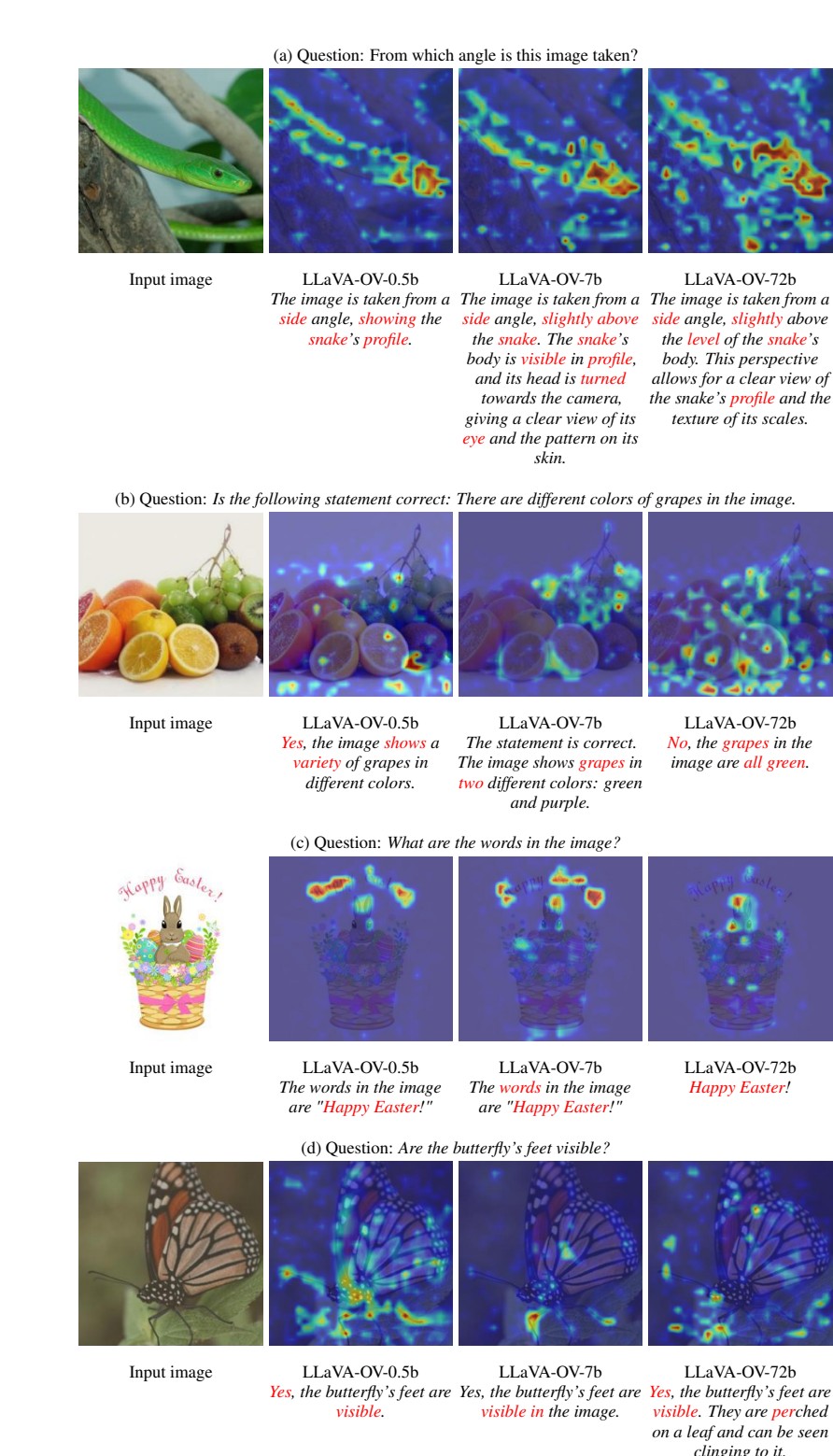

Figure 10: Comparison of the responses and focus region of LLaVA-OV with different LLM scales. The tokens in red denote the selected crucial tokens. The responses of the models with different scales often have different expressions, but the corresponding focus regions are often similar.

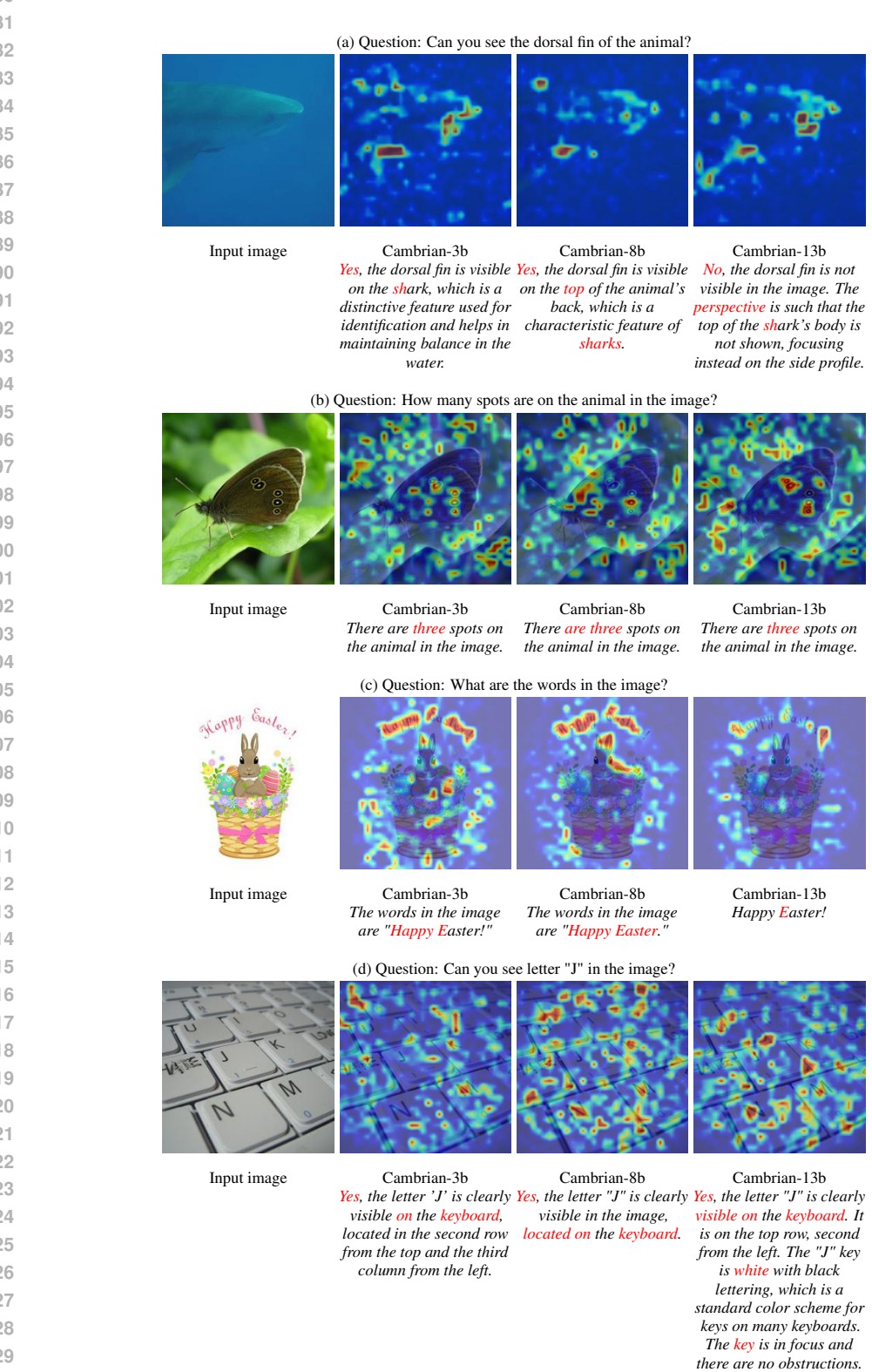

Figure 11: Comparison of the responses and focus region of Cambrian with different LLM scales. The tokens in red denote the selected crucial tokens. Cambrian often tend to attend to the whole image for more comprehensive visual understanding.

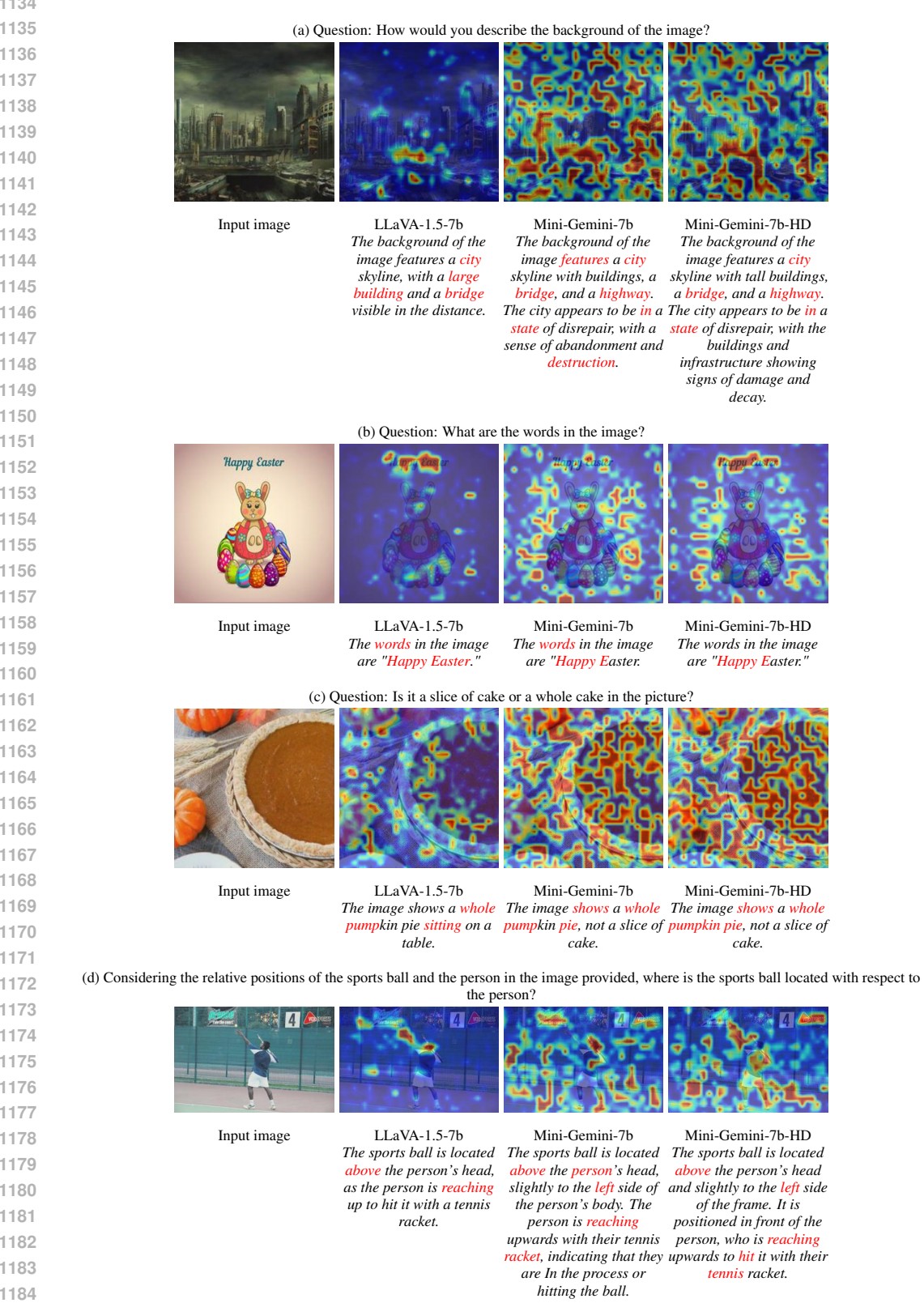

Figure 12: Comparison of the focus region of LVLMs with different vision architectures but using the same LLM. LLaVA-1.5-7b, Mini-Gemini-7b and Mini-Gemini-7b-HD all use Vicuna-1.5-7b as the LLM backbone. The tokens in red denote the selected crucial tokens. *HD* denotes the high-resolution vision encoder.

## A.10 HEATMAP VISUALIZATION OF QWEN2.5-VL AND QWEN3-VL

In this section we provide some heatmap visualization results of Qwen2.5-VL Bai et al. (2025) and Qwen3-VL, which adopt dynamic resolution in the vision encoder. As shown in Figure 13 and Figure 14, Qwen-VL family has relatively accurate visual attention. When the question can be answered with specific regions of the image, it can precisely focus on the details (examples (a)(b)(c) in Figure 13 and examples (a)(b) in Figure 14). When the question involves broader region of the image, it tends to enlarge the perception region to gain a more comprehensive visual understanding (examples (d)(e)(f) in Figure 13 and examples(c)(d) in Figure 14). These patterns may benefit from the dynamic resolution strategy that improves the visual perception and grounding.

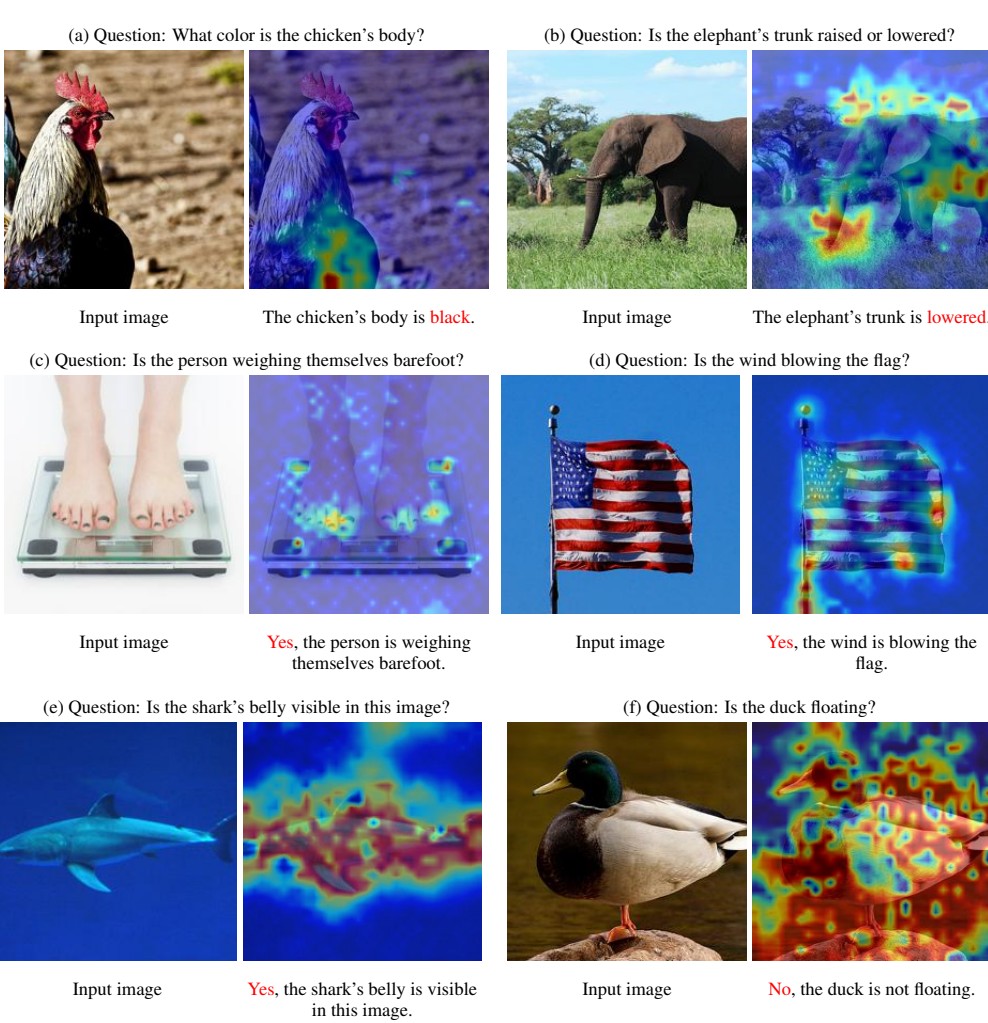

Figure 13: The heatmap visualizaion of Qwen2.5-VL.

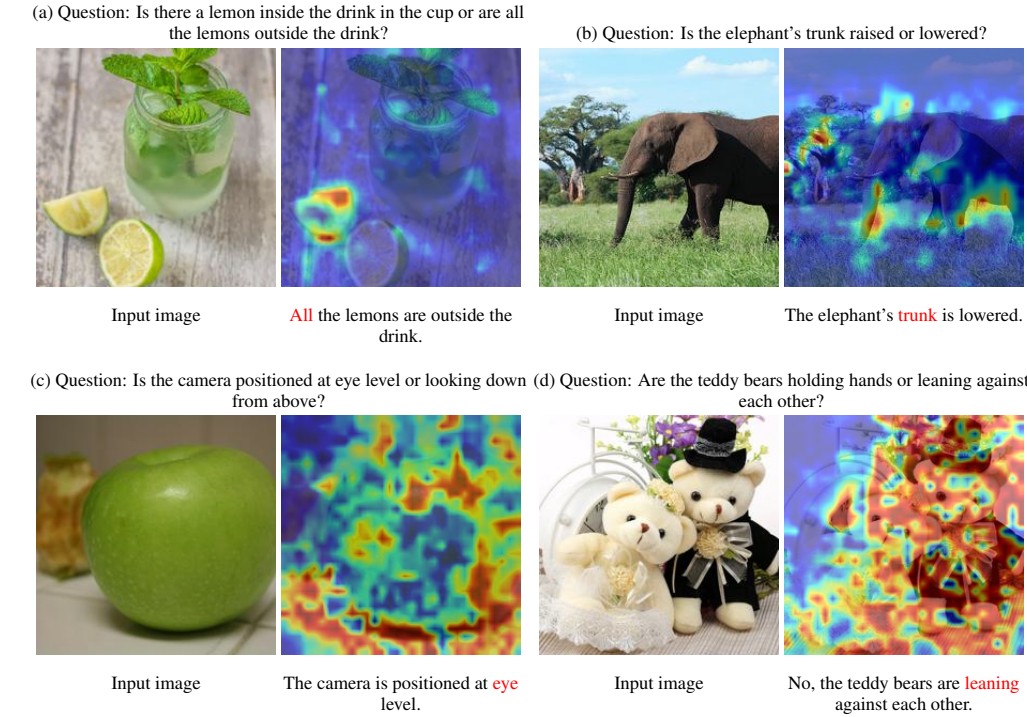

(a) Question: Is there a lemon inside the drink in the cup or are all the lemons outside the drink?

Input image — All the lemons are outside the drink.

(b) Question: Is the elephant's trunk raised or lowered?

Input image — The elephant's trunk is lowered.

(c) Question: Is the camera positioned at eye level or looking down from above?

Input image — The camera is positioned at eye level.

(d) Question: Are the teddy bears holding hands or leaning against each other?

Input image — No, the teddy bears are leaning against each other.

Figure 14: The heatmap visualizaion of Qwen3-VL.

## A.11 HEATMAP ASSOCIATED WITH HALLUCINATION

We provide some example results of the heatmap visualization on HallusionBench Guan et al. (2024) in Figure 15. HallusionBench is designed to evaluate both language hallucination and visual illusion. For each group of compared results, it contains the original image and question, counterfactual question (*i.e.,* replacing some keywords in the question that leads to different answers) and counterfactual image (*i.e.,* replacing some regions in the image that leads to different answers). In these examples, the LVLM consistently produces same answers under different variants and exhibit hallucination problem. In the first row, the model mainly focuses on the bowl of noodles at the bottom of the image, and pays low attention at the text. In the second and third rows, the model indeed looks at the relevant regions, but still fails to answer the questions. These examples provide two different causes of hallucinations, which can not be directly observed from the accuracy-based metrics. With the heatmap visualization, we can better diagnose the cause of hallucination and solve the problem accordingly.

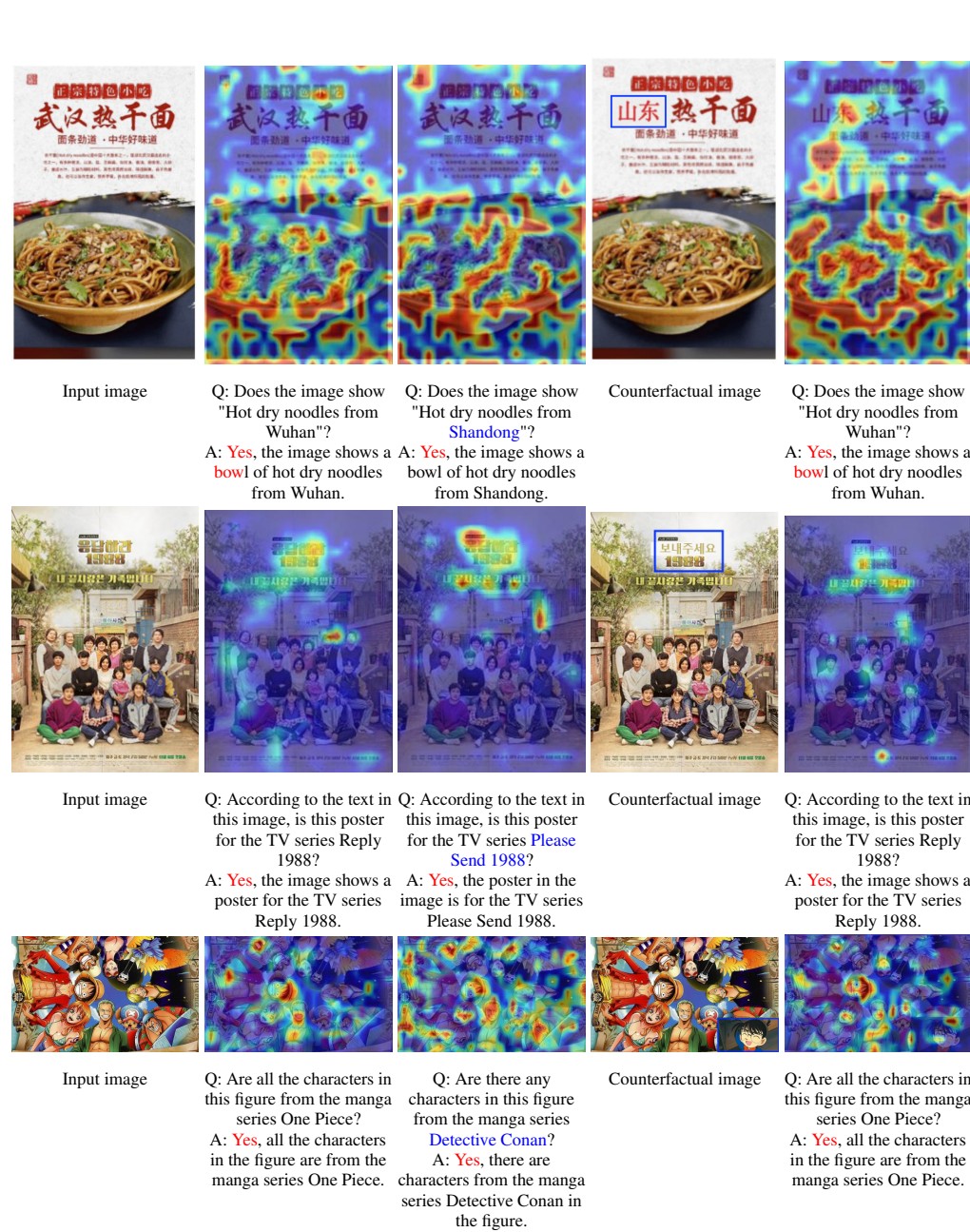

Figure 15: Examples of heatmap visualization on hallucinated responses on HallusionBench. Each group includes the original image and question, counterfactual question (the replaced words are highlighted in blue) and counterfactual image (the replaced regions are highlighted in blue boxes). The model may consistently give the same answer given counterfactual question or image.

