# OpenReview forum: "Where do Large Vision-Language Models Look at when Answering Questions?"
_ICLR.cc/2026/Conference — Submitted to ICLR 2026_

### Official Review · Reviewer_b64Z · 2025-10-26

**Soundness:** 3
**Presentation:** 2
**Contribution:** 2
**Rating:** 6
**Confidence:** 3

**Summary:**

The paper introduces a general, architecture-aware interpretability framework for LVLMs generation. It first identifies vision-relevant tokens in the generated answer via a log-likelihood ratio between the real image and a blurred “no-vision” baseline, then optimizes a single image-space mask for heatmap. Evaluated with insertion/deletion metrics across several benchmarks and model families, the method yields stronger, more faithful attributions and reveals several interesting insights.

**Strengths:**

1. Empirical study is thorough. Strong baselines, clearly defined insertion/deletion metrics, and targeted ablations that convincingly attribute gains to the proposed components.

2. Architecture-aware methodology. Unified pre-encoder masking and differentiable multi-resolution handling, with a single image-space iGOS++ mask that works across LVLM variants.

3. The insights are quite provocative with research motivation and questions clearly defined and solved. Pattern analysis among different LVLMs are also discussed.

**Weaknesses:**

> Experiments

The paper relies on a single metric family (insertion/deletion). Please clarify its robustness—in particular, sensitivity to sampling, baseline type, etc. Thus, Table 2 should report mean and standard deviation to prove superiority.

In addition, is this evaluation metric causal? By this I mean if it's possible to add some experiments to show what parts are actually causally emphasized in LVLM generation?

All current comparative baselines treat the LVLM as a white box. It would strengthen the evaluation to include black-box, perturbation-based baselines (e.g., D-RISE [1])

[1] Petsiuk, Vitali, et al. "Black-box explanation of object detectors via saliency maps." Proceedings of the IEEE/CVF conference on computer vision and pattern recognition. 2021.

**Questions:**

1. Why the result in Appendix A.2 is not aligned with human attention? It seems not intuitive.

2. Given your finding that visual grounding does not improve with larger LLMs, what method can solve this issue?

---

> ### Author Response · Authors · 2025-11-25
>
> We appreciate the recognition of our (1) thorough empirical study with strong baselines, clearly defined metrics, and targeted ablations, (2) architecture-aware methodology handling diverse LVLM architectures, (3) provocative insights and well-motivated research questions on analyzing the patterns across different LVLM families. For your main concerns and questions, we respond to them as follows:
>
> W-1. **[Metric robustness and causality]**:
>
> Thank you for the insightful comment. Deletion and insertion metrics [1][2] are widely utilized **causal** evaluation metrics of explanation algorithms. because they directly evaluate the model output on  perturbed images by removing or inserting the identified important regions. These metrics are causal: if masking out or revealing a region significantly alters the model’s prediction, that region is considered causally important for the model’s decision. In the experiments of Table 2, an insertion score of 0.8 means revealing about 20% of the most highlighted part of the image by the heatmap leads to the model generating the original sentence as if the full image is shown. Similarly, a deletion score of 0.2 means removing about 20% of the most highlighted pixels will lead to the model generating results as if the information from the image is all removed. Besides, upon the request of reviewer m8ag, we computed another metric that directly measures the accuracy of the generated sentence using LLMJudge. Namely, we utilized LLMJudge to judge whether the generated long sentence is consistent with the ground truth answer. And then we measured the accuracy of the model given 0%, 25%, 50%, 75%, 100% of the image, with the revealed pixels sorted by the importance given by the explanation heatmap. Instead of computing the area under the curve as the insertion metric, we can directly measure the accuracy drop under those scenarios on the MMVP dataset:
> | Method   | 0%   | 25%  | 50%  | 75%  | 100% |
> |----|----|---|------|------|------|
> | Grad-CAM | 0.14 | 0.28 | 0.35 | 0.40 | 0.50 |
> | iGOS++   | 0.14 | 0.42 | 0.46 | 0.50 | 0.50 |
>
> It can be observed that at each percentage of revealed pixels to the model, iGOS++ consistently achieves significantly higher accuracy than GradCAM. Besides, iGOS++ obtains accuracy close to the full image when only showing 25% of the image that are highlighted by the heatmap, which is better than showing 75% of the image with pixels Grad-CAM deemed important. In the final version we will provide this metric on all the datasets.
>
> To better clarify the robustness of the metrics, we conduct additional experiments to calculate the standard deviation. We re-run the experiments of LLaVA-OV on MMVP for 3 times and the results are shown in the table:
> |           | Exp.1 | Exp.2 | Exp.3 | avg | std |
> |-----------|-------|-------|-------|----|--|
> | Deletion  |  0.302   |   0.287  |  0.303  |   0.297  | 0.0073|
> | Insertion |    0.774  |   0.778  | 0.781  |  0.778  | 0.0029 |
>
> as one can see, the standard deviation is small and the improvements we’ve shown are significant with an unpaired t-test.
>
> W-2. **[Perturbation-based baselines]**:
>
> One of our baselines iGOS++ is also a perturbation-based interpretation method that treats the LVLM as a black box (only uses gradient w.r.t. the input, not like other heatmaps that required intermediate activations), and belongs to the same family with D-RISE. Both of them derive the saliency map of model prediction by introducing perturbation to the input image and measure the influence on the output.
> However, the main strategy of D-RISE formulation was specifically designed for object detection models and evaluates saliency by measuring similarity between detection vectors after perturbation (e.g., IoU between predicted bounding boxes), which does not directly transfer to open-ended LVLM generation. In object detection, [3] has shown that iGOS++ is comparable or better than D-RISE and runs 2-4 times faster. RISE-style approaches would be more preferable when the gradient w.r.t. the input is not available, but not advantageous in the current tested models where the gradient is available, especially since they are very slow. We can add RISE results in the final version.
>
> Importantly, D-RISE also adopts the deletion and insertion scores as its main evaluation metrics.
> We will include a more thorough discussion about the previous interpretation methods in the final version.
>
> [1] V Petsiuk. “Rise: Randomized input sampling for explanation of black-box models.” BMVC 2018
>
> [2] Petsiuk, Vitali, et al. "Black-box explanation of object detectors via saliency maps.". CVPR 2021
>
> [3] Jiang et al. “Diverse Explanations for Object Detectors with Nesterov-Accelerated iGOS++.” BMVC 2023

---

> > ### Author Response · Authors · 2025-11-25
> >
> > Q-1. **[Alignment with human attention]**:
> >
> > Thank you for inquiring about this important point. The misalignment between human attention and model saliency heatmap has been a widely discussed phenomenon. Heatmap visualization reflects the intrinsic behavior of LVLMs. The model attention is causal w.r.t. the model that may not align with human attention/behavior, as the LVLMs are not trained with human attention data. A potential reason is that LVLMs do not solve tasks using the same strategy as humans. Human vision tends to focus on semantically meaningful objects, while LVLMs may rely on low-level textures, spurious shortcuts learned from training data, or even **language bias** without visual information.
> >
> > Similar observations on VQA models have been discussed in several previous works, such as [1,3], and several benchmarks on visual grounding have been proposed, such as [4]. Many efforts are made to improve the **visual grounding** performance of vision models and VLMs, to ensure they derive the answer based on reliable visual attention. Some works like [1,2] have also explored incorporating human attention as supervision to guide the model attention.
> > Therefore, the model focus region interpretation and misalignment seen in Appendix A.2 is an important diagnostic signal, indicating reliance on spurious language bias and gaps between grounded visual reasoning. It is crucial to understand the intrinsic behavior of LVLMs to achieve targeted improvement, and to obtain some insights beyond accuracy-based performance, which motivates our work at model interpretation.
> >
> > Q-2. **[Improving visual grounding]**:
> >
> > From our investigations, the vision encoder architecture has a much stronger impact on the visual grounding behaviors compared with the scale of the LLM backbone, as shown in Table 4 and Figure 4 in the paper. For example, LLaVA-OV adopts a multi-resolution structure and shows more concrete visual attention on specific regions. In contrast, Cambrian integrates the visual features from multiple vision encoders and exhibits more comprehensive attention on the whole image. To adjust the visual attention behavior of LVLMs, potential factors from both our analysis and prior works include vision architecture design (e.g., spatial modeling, multi-resolution, multi-encoders,), training strategy (e.g., grounding-aware supervision, attention alignment, counterfactual data construction), vision-language fusion strategy (e.g., cross-attention fusion, multi-stage fusion, spatial-aware fusion). Our findings imply that the scaling of the LLM size may not solve all the problems, especially in terms of visual behaviors on multimodal tasks. We will add this discussion to the paper.
> >
> >
> > [1] Das et al. “Human Attention in Visual Question Answering: Do Humans and Deep Networks Look at the Same Regions?” EMNLP 2016.
> >
> > [2] Sood et al. “Multimodal integration of human-like attention in visual question answering.” CVPRW 2023
> >
> > [3] Yan et al. “Voila-A: Aligning Vision-Language Models with User’s Gaze Attention.” NeurIPS 2024
> >
> > [4] Xu et al. “MC-Bench: A Benchmark for Multi-Context Visual Grounding in the Era of MLLMs." ICCV 2025

---

### Official Review · Reviewer_m8ag · 2025-10-30

**Soundness:** 2
**Presentation:** 3
**Contribution:** 2
**Rating:** 2
**Confidence:** 5

**Summary:**

This paper adapts existing decision localization/visualization methods (GradCAM, iGOS++) for use with large vision-language models (LVLMs) in open-ended visual question answering, and leverages them to study how LVLMs attend to images when answering questions. Since these methods require a single prediction score, the paper proposes a token selection strategy which selects the tokens with high loglikelihood ratio (normalized by their loglikelihood given an neutral input image), and then computes the sum of the loglikelihood of these selected tokens as the prediction score. The paper also introduces a regularization term to the optimization objective of iGOS++. Several experiments are provided to compare the different decision localization methods, and explore how well LVLMs look at the correct regions in images.

**Strengths:**

This paper adapts existing decision localization/visualization methods (GradCAM, iGOS++) for use with large vision-language models (LVLMs) in open-ended visual question answering, and leverages them to study how LVLMs attend to images when answering questions. Since these methods require a single prediction score, the paper proposes a token selection strategy which selects the tokens with high loglikelihood ratio (normalized by their loglikelihood given a neutral input image), and then computes the sum of the loglikelihoods of these selected tokens as the prediction score. The paper also introduces a regularization term to the optimization objective of iGOS++. Several experiments are provided to compare the different decision localization methods and explore how well LVLMs look at the correct regions in images.

**Weaknesses:**

1. The comparison in Table 2 (and consequently the choice of iGOS++ and deletion/insertion metrics for the rest of the experiments) is incorrect. The problem is that iGOS++ directly optimizes the deletion/insertion metrics on test images, and then compares these values to methods that do not have this privilege (Grad-CAM etc). Note that directly optimizing any metric on test images will result in better values for that metric, but what actually matters is whether any generalization can be achieved beyond the optimized metric. For example, the paper's proposed optimization for the insertion metric could easily discover adversarial patterns on any test image that when inserted on a blurry background image will boost the likelihood of the LLM's prediction. In such cases, the "adversarial" interpretation would be of little use/meaning despite the high selection metric score, and so the paper must show improvement on an independent metric (for example object localization metrics, or overall accuracy as in [1]).

2. The paper misses several very relevant published papers in its related works. [1,2] have studied whether LVLMs attend to images. [3,4] have provided methods for decision localization/visualization in LVLMs.

3. The paper’s main findings are already established in previous work (“Q1: Do LVLMs rely on the input image when answering visual questions?” is answered in papers [1,2,3] and “Q2: Where do different LVLMs attend when generating variable-length responses? Q3: What is the relationship between answer correctness and focus region?“ is answered in [3,4]), so the paper’s novelty is limited to its proposed token selection method.

4. The paper provides no statistical confidence intervals for the metrics, so it is unclear how meaningful the small differences are. Some p-values are reported in the last section (lines 470-473), but the paper does not discuss how these values are computed (what statistical test, sample size, etc).

[1] The Instinctive Bias: Spurious Images lead to Illusion in MLLMs. EMNLP 2024.

[2] Unveiling the Ignorance of MLLMs: Seeing Clearly, Answering Incorrectly. CVPR 2025.

[3] MLLMs Know Where to Look: Training-Free Perception of Small Visual Details with Multimodal LLMs. ICLR 2025.

[4] V*: Guided Visual Search as A Core Mechanism in Multimodal LLMs. CVPR 2024.

**Questions:**

1. I suggest using a simpler object detection or VQA metric for comparing and validating the correctness of the localization methods.
2. Clarify novelty compared to several related published papers mentioned in the weaknesses section.

---

> ### Author Response · Authors · 2025-11-25
>
> W-1. **Evaluation metric**
>
> Thanks for your comments. First, we do not believe optimizing for the correct metric itself is necessarily an “unfair advantage”. If this logic of the reviewer stands, then any segmentation algorithm that optimizes the IoU metric [1] directly should be disqualified, since segmentations are usually evaluated with metrics built on IoUs. However those algorithms are accepted and widely used by the community because they improve the performance on the same important metric (IoU). Throughout the history of computer vision, an algorithm is  generally considered better if it directly optimizes an important metric for the task, not the other way as the reviewer suggested.
>
> In explanation, we argue that the insertion-deletion metrics are important metrics, as they are causal metrics that directly measure the change of predictions of the deep network when the input has been perturbed by pixels indicated by the heatmap that are attempting to explain it. This is significantly better than localization-based metrics, which a lot of earlier explanation methods utilize and led to methods obtaining a high score on those localization metrics by just generating significant object boundaries (hence are by definition within the bounding box of salient objects) without explaining the deep network (ln. 114-116 in the paper, see also [2], where those explanations still output the same object boundaries even when asked to explain an adversarial example classified as a completely different class). Hence, it has been long established in the explanation community that localization-based metrics are **not** faithful and shouldn’t be used.
>
> On the other hand, we do agree with the reviewer that the current computation of insertion-deletion metrics that only work on the likelihood of the generated sentence is not the only metric that can evaluate explanation algorithms. Based on the reviewer’s suggestion, we computed another metric in a similar vein to the insertion metric but more directly measured the “accuracy” as in the first reference the reviewer suggested. Namely, we utilized LLMJudge to judge whether the generated long sentence is consistent with the ground truth answer. And then we measured the accuracy of the model given 0%, 25%, 50%, 75%, 100% of the image, with the revealed pixels sorted by the importance given by the explanation heatmap. Instead of computing the area under the curve as the insertion metric, we can directly measure the accuracy drop under those scenarios on the MMVP dataset when partial images are shown:
> | Method   | 0%   | 25%  | 50%  | 75%  | 100% |
> |----------|------|------|------|------|------|
> | Grad-CAM | 0.14 | 0.28 | 0.35 | 0.40 | 0.50 |
> | iGOS++   | 0.14 | 0.42 | 0.46 | 0.50 | 0.50 |
>
> It can be observed that at each percentage of revealed pixels to the model, iGOS++ consistently achieves significantly higher accuracy than GradCAM. Besides, iGOS++ obtains accuracy close to the full image when only showing 25% of the image that are highlighted by the heatmap, which is even better than showing 75% of the image with pixels Grad-CAM deemed important. In the final version we will provide this metric on all the datasets.
>
> [1] H. Rezatofighi, et al. Generalized Intersection over Union: A Metric and A Loss for Bounding Box Regression. CVPR 2019
> [2] Nie et al. A Theoretical Explanation for Perplexing Behaviors of Backpropagation-based Visualizations. ICML 2018

---

> > ### Author Response · Authors · 2025-11-25
> >
> > W-2 & W-3. **Novelty w.r.t. missing related work**
> >
> > Thanks for pointing out those related works, we will cite them in the final version. We note that none of these related work focused on the same focus as our paper, which is locating the regions in the image that are the most important for the long sentences that LVLMs generate and make analyses based on those regions. Below we discuss each paper in detail.
> >
> > The first work provided insightful statistical and dataset analysis on how often LVLM use visual features, but did not provide a new approach to visualize salient regions that the LVLM are looking at.
> >
> > The second work compared LVLM’s attention between text and visual tokens as a whole, but didn’t differentiate among different visual tokens in their analysis and proposed several approaches to generate counterfactual text, whereas our work focuses on locating important visual regions the LVLM used.
> >
> > The third work located regions from the LVLM’s visual attention and cropped those as a new input to the LVLM, and their results showed that performance improved very significantly with the cropping strategy – which showed that LVLM’s decision-making is **highly inconsistent** with the regions their layers attended to. Differing from purely looking at attended regions, our method focuses on the regions that are actually important for the LVLM to generate the text, we believe that this provides different and complementary insights.
> >
> > The fourth work, similar to the third one, mainly proposed a search-based agent using LVLM to better locate visual details in high resolution images and constructed a benchmark to evaluate this capability. Their work focused on improving the capability of LVLMs to generate text on certain regions, but does not directly investigate on where the LVLMs are looking at when generating sentences.
> >
> > W-4. **Statistical evaluation for the metrics**:
> >
> > Thank you for raising this concern. We have conducted additional experiments to study the standard deviation of the metrics. We re-run the experiments of LLaVA-OV on MMVP for 3 times and the results are shown in the table:
> > |           | Exp.1 | Exp.2 | Exp.3 | avg | std |
> > |-----------|-------|-------|-------|----|--|
> > | Deletion  |  0.302   |   0.287  |  0.303  |   0.297  | 0.0073 |
> > | Insertion |    0.774  |   0.778  | 0.781  |  0.778  | 0.0029 |
> >
> > The standard deviation is small, which means our evaluation metric is stable across experiments.
> >
> > For the p-values reported in lines 470-473 in the paper, we conducted an independent t-test to evaluate whether the influence of the LLM size and vision encoder architecture are statistically significant to the deletion and insertion scores. The results indicate that the vision encoder structure has more significant influence on the focus regions. This experiment is conducted on the MMVP benchmark with 270 samples in total.

---

### Official Review · Reviewer_crB2 · 2025-10-30

**Soundness:** 3
**Presentation:** 3
**Contribution:** 2
**Rating:** 4
**Confidence:** 4

**Summary:**

The authors propose a method to extend traditional heatmap visualization techniques (e.g., GradCAM, iGOS++) to support Large Vision-Language Models (LVLMs) in open-ended visual question answering tasks. The key contribution is a visually relevant token selection mechanism that identifies tokens most dependent on image content through a log-likelihood ratio (LLR) between visual and blurred-image inputs. This approach enables the generation of interpretable visual heatmaps and facilitates analysis of how LVLMs attend to image regions when producing answers. The authors analyze multiple state-of-the-art LVLMs, like LLaVA-1.5, LLaVA-OneVision, and Cambrian across recent benchmarks including CV-Bench, MMStar, and MMVP.

**Strengths:**

1.	The paper is clearly written, well structured, and easy to follow.
2.	LVLM interpretability is a critical and underexplored topic, especially considering the success of models such as LLaVA, InstructBLIP, Gemini, and GPT-4o, and the concerns raised by benchmarks like MMVP, MERLIM, AMBER, POPE, and HallusionBench regarding hallucinations and insufficient visual grounding. The proposed tool can help diagnose whether such errors stem from limited grounding capacity or architectural constraints that prevent LVLMs from fully leveraging visual information.
3.	The approach effectively identifies which tokens rely on visual grounding and visualizes how this information is used via interpretable heatmaps. The paper includes several qualitative examples that illustrate the findings, supported by the corresponding mathematical derivations.
4.	The study extends and compares multiple heatmap-based interpretability methods, including GradCAM, T-MM, IIA, and iGOS++, providing a useful empirical reference for future LVLM analysis.

**Weaknesses:**

1.	My main concern is the lack of novelty. Although this paper addrees a relevant topic, the method primarily combines existing components (heatmap visualization + token selection) rather than introducing a fundamentally new paradigm. The LLR-based token selection is a natural extension of standard likelihood analysis, and while the integration is non-trivial, it could be seen as incremental.

**Questions:**

1. It would be insightful to analyze the visual attention (heatmap) associated with hallucinated or spurious tokens to better understand whether these hallucinations arise from incorrect grounding or over-reliance on textual priors. Could you provide such an analysis?
2.	The hyperparameters α and λ seem potentially architecture-dependent, but the paper does not report how they behave across different LVLMs. Have you studied their sensitivity or generalization across models?

---

> ### Author Response · Authors · 2025-11-25
>
> We appreciate the recognition of (1) clarity of paper writing, (2) significance in diagnosing hallucination problems of LVLMs, (3) effectiveness of visual interpretation, and (4) useful empirical reference for future LVLM analysis. To address the major concerns of the reviewer, we respond to each of them as follows:
>
> W-1. **[Novelty of this work]**:
>
> We appreciate your valuable comments on the contribution of this work. Although our method is built on top of existing visual interpretation methods, we want to clarify that we are the **first** to extend previous image classification interpretation methods to support free-form generation of open-ended responses by LVLMs. There has been no prior work on utilizing visual explainability methods to explain vision-language models with long-sentence generation in the image space, which has much greater complexity in both outputs and model structure. We also introduced some additional improvements to the existing interpretation methods to better adapt to LVLMs with complex architectures, as described in Section 3.3. In this regard, we consider the work to be novel and worth publishing, even as a baseline for future approaches.
>
> The LLR-based token selection we proposed is natural but effective, and aligns well with the autoregressive generation of LVLMs since both of them rely on conditional probability of next token prediction. Thanks to the autoregressive nature of LVLMs, the LLR term can be efficiently computed in a single forward pass. Furthermore, as shown in Table 3 of the paper, our token selection approach outperforms some intuitive methods like aggregating joint probability of the whole output or part-of-speech based keywords selection.
>
> Another important contribution of our work is providing visual behavior analysis of state-of-the-art LVLMs, uncovering architectural patterns in visual grounding and offering insights that are difficult to obtain from classification-based saliency methods or accuracy-based evaluation. We’ll clarify the contribution better in the final version.
>
> Q-1. **[Heatmap associated with hallucinated tokens]**:
>
> Thank you for the insightful comment. Analyzing the visual attention of hallucinated or spurious tokens is indeed an important direction. To answer this question, we conducted additional experiments on HallusionBench and show them in **Appendix A.11** of the revised manuscript. HallusionBench includes image-question pairs to evaluate both language hallucination and visual illusion. Given an original question and answer pair, it intentionally replaces the keyword or the essential region of the image that leads to different answers to the question. The results show that the LVLMs did notice the correct regions, but still gave the wrong answer. Such results can be better understood from the images in column 3 of Fig. 15 in the new appendix when the questions are perturbed. Despite looking at the correct regions, the LVLMs answered yes to all the counterfactual questions, showing that they may not understand the text in the images in the questions (Chinese characters “Shandong”, Korean characters “Reply” and an anime character in Conan). This shows that our method provides additional insights for those diagnoses, and users can use it with many different questions to probe the LVLM for diagnosing the actual reason for the hallucination.

---

> > ### Author Response · Authors · 2025-11-25
> >
> > Q-2. **[Sensitivity of α and λ]**:
> >
> > Thank you for raising this concern. We have conducted additional experiments to study the sensitivity of the parameters $\lambda_1, \lambda_2, \lambda_3, \alpha$ across different LVLMs. Specifically, we run experiments on different combinations of the parameters and show the results on LLaVA-1.5, LLaVA-OV and Qwen2.5-VL. The results are shown in the tables below. Extremely small or large values of the regularization terms may degrade the deletion and insertion scores. Small threshold $\alpha$ may select too many tokens and introduce noise, while large $\alpha$ causes the interpretation to miss relevant content in the output. Despite variance in the deletion and insertion scores, in general the interpretation results are not very sensitive to the parameters in a reasonable range and we can consistently derive meaningful heatmaps.
> >
> > | $\lambda_1$ | LLaVA-1.5 Del ↓ | Ins ↑ | LLaVA-OV Del ↓ | Ins ↑ |  Qwen2.5-VL Del ↓ | Ins ↑ |
> > |----|------:|------|----:|------|---:|---|
> > | 0.5 | 0.395 | 0.775 | 0.352 | 0.787 | 0.3529 | 0.5864
> > | 1.0 | 0.366 | 0.811 | 0.305 | 0.778 | 0.2844 | 0.5906
> > | 2.0 | 0.403 | 0.781 |0.357 | 0.787| 0.3761| 0.5669
> >
> > | $\lambda_2$ | LLaVA-1.5 Del ↓ | Ins ↑ | LLaVA-OV Del ↓ | Ins ↑ |  Qwen2.5-VL Del ↓ | Ins ↑ |
> > |----|------:|------|----:|------|---:|---|
> > | 0.0 |0.381 | 0.794 |0.360 |0.780 | 0.3982 | 0.5723
> > | 0.1 | 0.366 | 0.811 | 0.305 | 0.778 | 0.2844 | 0.5906
> > | 0.5 |0.403 |0.787  |0.382|0.785|0.3778| 0.6164
> >
> > | $\lambda_3$ | LLaVA-1.5 Del ↓ | Ins ↑ | LLaVA-OV Del ↓ | Ins ↑ |  Qwen2.5-VL Del ↓ | Ins ↑ |
> > |----|------:|------|----:|------|---:|---|
> > | 5.0 | 0.401 | 0.783 | 0.398| 0.788 |0.3512|0.5926
> > | 10.0 | 0.366 | 0.811 |0.305|0.778 | 0.2844 | 0.5906
> > | 15.0 | 0.404 | 0.783 |0.375 | 0.795|0.4270 | 0.6245
> >
> > | $\alpha$ | LLaVA-1.5 Del ↓ | Ins ↑ | LLaVA-OV Del ↓ | Ins ↑ |  Qwen2.5-VL Del ↓ | Ins ↑ |
> > |----|------:|------|----:|------|---:|---|
> > | 0.5 | 0.414 | 0.774 | 0.388 | 0.761|0.2801 | 0.5950
> > | 1.0 | 0.366 | 0.811 |0.305 | 0.778 |0.2844|0.5906
> > | 2.0 | 0.425 | 0.792 |0.432 | 0.781 | 0.3999 | 0.6037

---

### Official Review · Reviewer_eJZt · 2025-11-01

**Soundness:** 3
**Presentation:** 3
**Contribution:** 2
**Rating:** 6
**Confidence:** 2

**Summary:**

This paper dives into a key question about large vision-language models: where do they focus on images when answering questions? Since existing heatmap visualization methods mostly work for classification tasks, the authors extended these tools to handle VLMs' variable-length, autoregressive outputs. The authors propose a method to identify "visually relevant tokens" from model responses. Using this method, they tested state-of-the-art LVLMs like LLaVA-1.5, LLaVA-OV, and Cambrian on vision-centric datasets. The results showed that these models do rely on images for over 75% of cases, different vision architectures lead to distinct attention patterns, and simply making the LLM bigger doesn’t change how the model looks at images much. They also found that even when models get answers wrong, their focus regions often relate to the question, and sometimes, correct answers come from irrelevant image parts. These findings further highlight generalization challenges.

**Strengths:**

1. The proposed method fixes a gap for interpreting VLMs at the sentence level by filtering out the language stuff that doesn’t need images; the heatmaps make way more sense.

2. The authors conduct experiments on different LVLMs with different setups and multiple vision-heavy datasets. These experiments make their conclusion more reliable.

3. The generalization of the proposed method works with different heatmap tools. This makes it easy for others to follow.

**Weaknesses:**

1. Although the authors provide several models in the experiment part, they lack the results on the frontier open-source model, like Qwen3-VL. Showing that these models maintain the same property as previous models is important for others to understand these models. Also, results on larger models should be discussed.

2. The authors mainly provide results on some perception benchmarks. I would also like to know the phenomenon of these models on some reasoning benchmarks and high-resolution perception benchmarks. Incorporating these results will strengthen the overall paper.

3. More analysis of visual encoder should be included. New visual encoders do not use fixed resolution, like in Qwen2-VL and newer model. Will these new models perform differently as they improve visual perception?

**Questions:**

As stated in the weakness part.

---

> ### Author Response · Authors · 2025-11-24
>
> We sincerely thank the reviewer for recognizing our work valuable in (1) addressing the gap in sentence-level interpretation of VLMs and achieving more reasonable heatmaps, (2) providing extensive experiments across diverse LVLMs and datasets, and (3) having strong generalization across different interpretation methods. Regarding  the main concerns of the reviewer, we address them as follows:
>
> W-1.**[Results on Qwen-VL]**:
>
> We thank the reviewer for pointing this out. Following your suggestion, we have additionally conducted experiments on Qwen3-VL and Qwen2.5-VL and show the heatmap visualization in **Appendix A.10** of the revised manuscript. The resulting deletion and insertion scores on the MMVP dataset are shown in the table below.
> |    | Deletion ↓ | Insertion ↑ |
> |---|----|----|
> | Qwen2.5-VL | 0.277  | 0.641   |
> | Qwen3-VL | 0.436 | 0.671 |
>
> These results are similar to the ones from Cambrian on this dataset. Qwen2.5-VL and Qwen3-VL adopt a similar dynamic resolution strategy, and both of them have relatively accurate visual attention. When the question can be answered with specific regions of the image, it can precisely focus on the details. When the question involves a broader region of the image, it tends to enlarge the perception region to gain a more comprehensive visual understanding. These patterns may benefit from the dynamic resolution strategy that improves the visual perception and grounding. Results showed that Qwen3-VL may be more robust to occlusion as it is harder to drop the Qwen3-VL score by masking regions. Due to the fact that Qwen3-VL only provided an API without the ability to retrieve intermediate information, we couldn’t figure out a way to run the baselines that rely on intermediate activations, which showed the general appeal of our method (only requiring the gradient to the input).
>
> **[Results on larger models]**:
> As shown in Table 4 in the paper, we have evaluated LVLMs spanning a wide range of parameter scales—from **0.5B to 72B**. Our main observation is that given similar vision-encoder architectures, the language model size has no significant impact on the visual focus region produced by the model. In contrast, the vision architecture has greater influence on the visual attention behavior of LVLMs.
>
>
> W-2. **[Reasoning and high-resolution benchmarks]**:
>
> Thank you for the valuable feedback. We’d like to note that we have already evaluated on datasets containing  reasoning questions and questions about high-resolution and detail-dependent images. Question types of the evaluated datasets are as follows:
>
> **MMStar** evaluates 6 core capabilities: Fine-grained Perception, Coarse Perception, Instance Reasoning, Logical Reasoning, Mathematics, Science & Technology, where each category can be further separated into 3 detailed axes. **CV-Bench** mainly evaluates spatial relationship reasoning and counting. **LLaVA-Bench** contains questions about conversation, detailed description, complex reasoning, as well as more challenging tasks including indoor and outdoor scenes, memes, paintings, sketches, etc.
>
> Therefore, our experiments **already covered** a broad set of reasoning-intensive questions, and questions about fine-grained objects attributes requiring detailed perception. Furthermore, in Figure 5 we separate the questions into 5 types: Spatial, Attributes, Counting, Global Understanding and Reasoning, and studied the relationship between answer correctness and focus region plausibility across these categories. The results indicate that LVLMs generally attend to the relevant regions when answering reasoning and detailed attribute questions (which is not always consistent for other question types). However, the answer accuracy on questions about detailed attributes is higher than reasoning questions, suggesting improving performance on visual reasoning tasks may require strengthening the model’s high-level reasoning ability rather than its visual grounding capability.

---

> > ### Author Response · Authors · 2025-11-24
> >
> > W-3. **[Analysis of Visual Encoder]**:
> >
> > We appreciate the reviewer’s valuable comment. Our work already includes analyses across different visual encoder architectures, including fixed-resolution models (LLaVA-1.5), multi-encoder systems (Cambrian), multi-resolution models (LLaVA-OV), and high-resolution models (Mini-Gemini-HD). These models cover the major design paradigms of current LVLMs. In Section 4.5, we provide empirical analysis of their visual attention behaviors and discuss how these behaviors relate to the underlying encoder architecture. For example, LLaVA-OV tends to produce disjunctive, highly localized attention on specific visual patches which aligns with the multi-resolution encoder design to extract detailed features. Cambrian and Mini-Gemini-HD show more scattered and compositional attention spread across the image, as they adopt multiple vision encoders that may attend to different image regions.
> >
> > To further address the reviewer’s question on newer LVLMs, we have additionally conducted experiments on Qwen3-VL and Qwen2.5-VL, which introduces dynamic resolution processing. And we have included the visualized heatmaps in **Appendix A.10** of the revised manuscript. Please refer to the discussion in **W-1 [Results on Qwen-VL]**.

---

> > > ### Comment · Reviewer_eJZt · 2025-11-25
> > >
> > > I appreciate the author's effort for the rebuttal. However, as the models mentioned in the original paper mainly use fixed resolution encoder, even if they claim they support high resolution image analysis, they still use the dynamic-partition strategy when dealing with high resolution images. This means that the visual encoder still tackles a set of fixed resolution images and then cat them to recover the original high resolution image. However, for recent visual encoders like Qwen2-VL(Qwen-ViT), they support native image perception and make some changes to the visual encoder, so my primary concern is whether the method in the paper still work for this kind of new visual encoder.

---

> > > > ### Author Response · Authors · 2025-11-26
> > > >
> > > > We thank the reviewer for the follow-up question. We would like to emphasize that our method does not rely on any assumption about how the visual encoder partitions or processes the input image. In our algorithm, the saliency mask is at a lower resolution than the image (e.g., 32x32), when optimizing, it is upsampled to the same size as the input image. When there’re multiple resolutions, the same mask is upsampled to different resolutions and the masked images are sent through the LVLM’s visual encoder without modifying or interacting with the encoder’s internal mechanism.
> > > >
> > > > Therefore, no matter how the visual encoder processes the input image, whether fixed-resolution, dynamic resolution or multi-scale, it processes the masked image using exactly the **same** process.
> > > >
> > > > Our heatmap visualization method inherently adapts to any visual encoder architecture. In the rebuttal we have also conducted experiments on Qwen2.5-VL and Qwen3-VL, as shown in **Appendix A.10** of the revised manuscript , demonstrating that our method continues to work as expected on encoders with native dynamic resolution.

---

### Author Response · Authors · 2025-11-25

We thank all reviewers for the comments and suggestions. We have revised the paper to address some concerns, with updates highlighted in blue in Appendix A.10 and A.11.

Specifically, we added in the revised manuscript:
- Heatmap visualization and analysis of Qwen2.5-VL and Qwen3-VL (A.10)
- Study of the focus region related to hallucinated answers on HallusionBench (A.11)

Besides, we also added a new accuracy-based metric as suggested by reviewer m8ag, which compares the answer accuracies when given 0%, 25%, 50%, 75%, 100% of the image, with the revealed pixels sorted by the importance given by the explanation heatmap.

We hope these revisions and our responses adequately address the reviewers’ concerns and look forward to replies and further discussion.

---

### Comment · Area_Chair_uYiM · 2025-11-25
**Discussion Period**

Dear Reviewers and Authors,

Thank you to the authors for submitting your rebuttal. We kindly encourage reviewers to take a moment to read the response and share any follow-up thoughts. Your timely engagement at this stage is highly valuable and helps ensure a fair, well-informed final decision.

We appreciate everyone’s efforts and contributions to the process.

Warm regards,
Your AC

---

### Meta-Review · Area_Chair_qhkg · 2026-01-08

**Summary:**

This paper received scores of 6, 4, 2, 6.  Initial concerns include lacking results of frontier open-source models eg Qwen3-VL, lacking reasoning benchmarks and high-resolution perception benchmarks, more analysis of visual encoder needed, lacking novelty in particular combining existing components (heatmap visualization + token selection) rather than introducing a fundamentally new paradigm, missing closely related work, incorrect comparisons/analyses, no statistical confidence intervals for metrics, and lacking black-box, perturbation-based baselines.

**Reviewer Concerns:**

Many of the original reviews were addressed by the rebuttal.  However, two main concerns remain. The biggest concern is regarding incremental novelty, raised by multiple reviewers. Specifically, the core method largely combines existing components (perturbation-based heatmap visualization and likelihood-based token selection) without introducing a fundamentally new paradigm. The second concern is the issue of problematic comparisons in Table 2 is not fully addressed. The rebuttal addresses the strongest form of the criticism, but the extent to which methodological differences affect the reported comparisons remains somewhat unclear.  Due to these concerns, the paper does not meet the bar for acceptance.

**Reviewer Scores:**

Based on the rebuttal, it is possible that the reviewers may have kept their original scores especially due to the novelty concerns.

---

### Decision · Program_Chairs · 2026-01-26

Reject